# A nonparametric method for gradual change problems with statistical guarantees

**Lizhen Nie**
Department of Statistics
The University of Chicago
lizhen@statistics.uchicago.edu

**Dan Nicolae**
Department of Statistics
The University of Chicago
nicolae@statistics.uchicago.edu

## Abstract

We consider the detection and localization of *gradual* changes in the distribution of a sequence of time-ordered observations. Existing literature focuses mostly on the simpler *abrupt* setting which assumes a discontinuity jump in distribution, and is unrealistic for some applied settings. We propose a general method for detecting and localizing gradual changes that does not require a specific data generating model, a particular data type, or prior knowledge about which features of the distribution are subject to change. Despite relaxed assumptions, the proposed method possesses proven theoretical guarantees for both detection and localization.

## 1  Introduction

In a sequence of time-ordered observations $\{Y_{t,T} : t = 1, 2, \cdots, T\}$, the aim of change point detection (CPD) is to (a) detect: answer the question of *whether* the distribution of $Y_{t,T}$ changes, and (b) localize: if it changes, answer the question of *when*. The classic formulation of CPD usually assumes that the possible change point is *abrupt*, i.e., there is a discontinuity jump in the distribution of $Y_{t,T}$, leading to a simpler problem. However, in many real-life situations, the changes in a sequence happen *smoothly* or *gradually*, rather than abruptly. Figure 1 illustrates some examples.

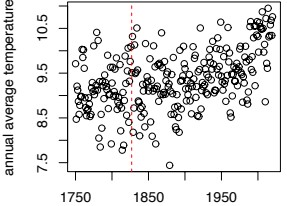
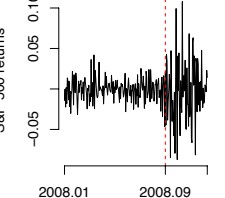
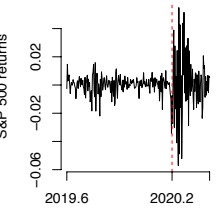

(a) Annual average temperature in central England.   (b) S&P 500 stock index daily returns.

Figure 1: Examples of gradual changes. The vertical red dashed lines indicate the gradual change start points estimated by the method proposed in this paper.

The first example concerns climatology, and investigates the temperature patterns over years. Figure 1a depicts the annual average temperature in central England from 1750 to 2020, where we observe a smooth increase starting around 1850. The second example comes from finance. The S&P 500 stock index is an important indicator of the overall market. As shown in Figure 1b, its volatility level usually remains constant in a stable market, and then gradually increases with the development of some events such as the financial crisis in 2008 or the COVID-19 pandemic in 2020.

Despite the wide variety of applications, inference for gradual changes is under-researched, and most existing methods require domain knowledge. Early research assumed that the gradual change follows a particular parametric model. For example, Lombard (1987) considers a setting where some

unknown parameter changes linearly, while others (Hušková, 1999; Hušková and Steinebach, 2002; Aue and Steinebach, 2002) consider models with polynomial changes.

Recent methods also consider nonparametric settings. However, most of them still require specific assumptions on the data model. For example, Muller (1992); Raimondo (1998); Goldenshluger et al. (2006) consider the location model where first order moment of observations changes. Mallik et al. (2011, 2013) investigate a stronger assumption: the mean change is monotonic. Mercurio et al. (2004) consider the volatility model where second order moment of observations fluctuates. Quessy (2019) assumes that the sequence follows two stationary distributions at the beginning and the end, and the changing phase in-between is a mixture of them with weights changing linearly with time.

As far as we know, Vogt and Dette (2015) is the only nonparametric method that applies to general types of models and data types. Despite its generality, the method proposed in Vogt and Dette (2015) requires prior knowledge about which stochastic feature(s) might change. Moreover, their method requires specification of a threshold determined through expensive simulations. Also, Vogt and Dette (2015) considers only the localization problem, while ignoring the detection step which is shown to be important for false positive control in real-data applications (Van den Burg and Williams, 2020).

We propose a nonparametric method for detecting and localizing gradual changes. The proposed method requires no prior domain knowledge, and we offer theoretical guarantees on both detection (false positive rate, power) and localization (consistency).

## 2  Problem Statement

Suppose we observe a time-ordered independent sequence $\{Y_{t,T} : t = 1, 2, \cdots, T\}$ taking values in a general metric space $(\mathcal{Y}, \|\cdot\|_{\mathcal{Y}})$. $Y_{t,T}$ is observed at time $u = t/T \in [0,1]$. We are concerned with:

1. (Detection) Deciding whether the distribution of observation changes with time $u$. This is formulated as a hypothesis testing problem with null $H_0$ and alternative $H_A$ hypotheses shown below. Let $P_u$ be a probability measure on $(\mathcal{Y}, \|\cdot\|_{\mathcal{Y}})$ such that $Y_{t,T} \sim P_u$ for $u = t/T$, then

$H_0 : P_u$ is constant over $u \in [0,1]$.

$H_A : P_u$ is constant over $u \in [0, u_0]$ for some $u_0 \in (0,1)$, but is not constant over $u \in [0,1]$. (1)

Further, we assume that $P_u$ is continuous with respect to the weak topology in the sense that $\forall u \in [0,1]$, $P_v$ weakly converges to $P_u$, as $v \to u$.

2. (Localization) If rejecting $H_0$ in step 1, obtain an estimator $\hat{\rho}$ of the gradual change point $\rho^*$ where the probability measure $P_u$ *starts* to change, i.e., $\rho^* := \sup\{u : P_v = P_0, \forall v \in [0, u]\}$.

Notice that we do not put specific assumptions on the data type or distribution of $Y_{t,T}$ and thus, our formulation allows a large number of special models such as

$$\text{location model:} \quad Y_{t,T} = \mu(t/T) + \varepsilon_t, \tag{2}$$
$$\text{volatility model:} \quad Y_{t,T} = \sigma(t/T)\varepsilon_t, \tag{3}$$

where $\mu(\cdot), \sigma(\cdot)$ can be any continuous function, and $\varepsilon_t$'s are zero mean i.i.d errors.

**Notations.** We denote $\lceil x \rceil$ as the least integer no smaller than $x$, $\mathbf{1}_d = (1, \cdots, 1)^\top \in \mathbb{R}^d$, $\boldsymbol{I}_d \in \mathbb{R}^{d \times d}$ the identity matrix. We use $\mathbb{I}$ to denote indicator function, $\xrightarrow{w}$ weak convergence, $\mathbb{Z}_+$ the set of positive integers. For a set of constants $a_T, b_T$ and random variables $X_T$, we write $a_T = \Theta(b_T)$ if there exist constants $C_1, C_2 > 0, t_0 \in \mathbb{Z}_+$ s.t. $C_1 a_T \leq b_T \leq C_2 a_T, \forall T \geq t_0$. Denote $X_T = O_p(a_T)$ if $X_T/a_T$ is stochastically bounded, and $X_T = o_p(a_T)$ if $X_T/a_T$ converges to zero in probability.

## 3  Methodology

**Existing statistic.** We consider first univariate $Y_{t,T}$'s. Suppose the change is in $\mathbb{E}Y_{t,T}$; traditional CUSUM statistic (Page, 1954) solves CPD problem by defining

$$\widehat{C}_T(u,v) = 1/T \sum_{t=1}^{\lceil vT \rceil} Y_{t,T} - v/(uT) \sum_{t=1}^{\lceil uT \rceil} Y_{t,T}, \quad \text{for any } 0 \leq v < u \leq 1.$$

which compares cumulative sums of $Y_{t,T}$ over different time spans $[0, v]$ and $[0, u]$. Then

$$\widehat{\mathcal{D}}_T^{\mathrm{uni}}(u) = \max_{v \in [0,u]} |\widehat{C}_T(u,v)|, \quad \text{for any } 0 \leq u \leq 1.$$

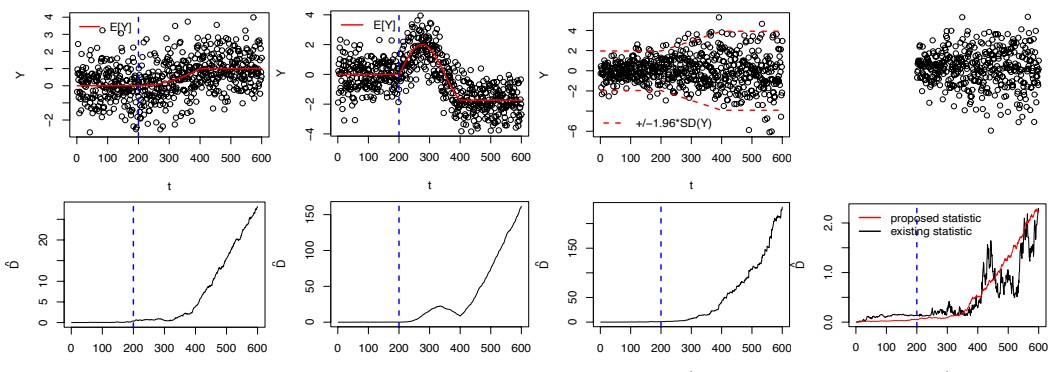

Figure 2: Plots of $Y_{t,T}$ (top row) and their $\widehat{\mathcal{D}}_T^{\text{gen}}(t/T)$ (bottom row) against $t$. The blue vertical line denotes true change point. Data in column 1, 2 follow location model (2) with $\varepsilon_t \sim N(0,1)$, and $\mu_1(u) = \mathbb{I}(1/3 \leq u \leq 2/3)(3u-1)^{1.5} + \mathbb{I}(u > 2/3)$, $\mu_2(u) = 2\sin(4\pi(u-1/3))\mathbb{I}(1/3 \leq u \leq 2/3) + 2\sin(4\pi/3)\mathbb{I}(u \geq 2/3)$, respectively. Data in column 3 follows volatility model (3) with $\varepsilon_t \sim N(0,1)$ and $\sigma(\cdot) = \mu_1(\cdot) + 1$. Column 1, 2 set $\mathcal{F} = \{f : x \mapsto x\}$, and column 3 $\mathcal{F} = \{f : x \mapsto x^2\}$.

can be used to detect changes in feature $\mathbb{E}Y_{t,T}$ over time span $[0, u]$. Intuitively, if there are no changes over $[0, u]$, $\widehat{\mathcal{D}}_T^{\text{uni}}(u)$ should be small. For example, in Figure 2, the first and second column depicts a sequence with change in $\mathbb{E}Y_{t,T}$ (shown in top row), and $\widehat{\mathcal{D}}_T^{\text{uni}}$ (shown in bottom row) take small values before $\tau^* = 200$ where $\tau^* = \lceil T\rho^* \rceil$, and then grow substantially. Thus, $\widehat{\mathcal{D}}_T^{\text{uni}}(u)$ essentially measures the variation over $[0, u]$ in these univariate settings.

For multivariate/structured $Y_{t,T}$ or for changes in more general features of the form $\mathbb{E}f(Y_{t,T})$ where $f : \mathcal{Y} \to \mathbb{R}$ is a measurable function, Vogt and Dette (2015) replaces $\widehat{\mathcal{D}}_T^{\text{uni}}$ with

$$\widehat{\mathcal{D}}_T^{\text{gen}}(u) = \sup_{f \in \mathcal{F}} \max_{v \in [0,u]} |\widehat{C}_T(u, v, f)|, \quad \text{where}$$
$$\widehat{C}_T(u, v, f) = 1/T \sum_{t=1}^{\lceil vT \rceil} f(Y_{t,T}) - v/(uT) \sum_{t=1}^{\lceil uT \rceil} f(Y_{t,T}). \tag{4}$$

$\widehat{\mathcal{D}}_T^{\text{gen}}$ takes supremum over a pre-specified set of functions $\mathcal{F}$ to ensure that changes in $\mathbb{E}f(Y_{t,T})$ for all $f \in \mathcal{F}$ are considered. Note that $\widehat{\mathcal{D}}_T^{\text{uni}}$ is a special case of $\widehat{\mathcal{D}}_T^{\text{gen}}$ with $\mathcal{F} = \{f : x \mapsto x\}$, and column 3 of Figure 2 sets $\mathcal{F} = \{f : x \mapsto x^2\}$.

There are three main issues with $\widehat{\mathcal{D}}_T^{\text{gen}}$. First, it relies heavily on the pre-specified function class $\mathcal{F}$. Also, to calculate $\widehat{\mathcal{D}}_T^{\text{gen}}$, $\mathcal{F}$ can only contain a finite (usually small) number of functions (e.g., $f : x \mapsto x$ or $f : x \mapsto x^2$), the choice of which relies heavily on prior knowledge about which features might change. When $\mathcal{F}$ is misspecified, $\widehat{\mathcal{D}}_T^{\text{gen}}$ can be non-informative and fail subsequent tasks. Second, $\widehat{\mathcal{D}}_T^{\text{gen}}$ does not consider the scale of $\widehat{C}_T(\cdot, \cdot, f)$ which could be incomparable for different $f$'s. Third, the limiting distribution of $\widehat{\mathcal{D}}_T^{\text{gen}}(\cdot)$ is unknown, leading to computational challenges in subsequent analyses.

**Proposed statistic.** We introduce a new statistic that puts minimal assumptions on data types and generating process, and is free of the issues discussed above. It is motivated by the recent success of applying kernel approaches to abrupt CPD problems (e.g., Harchaoui et al. (2008); Li et al. (2015); see Section 7 for more details). These kernel approaches assume access to a positive semidefinite kernel $k : \mathcal{Y} \times \mathcal{Y} \to \mathbb{R}$ that measures pairwise similarity among observations. Compared with features, kernels are more flexible and easier to specify, especially for structured data, showing great potential for solving gradual CPD problem. Inference starts with measuring data variation in time span $[0, u]$; for each possible change point $v < u$, $v$ divides the observations into two groups: those coming before $\lceil Tv \rceil$ and those after $\lceil Tv \rceil$. Note that the average similarity among observations within the same group is:

$$\widehat{S}_T^{\text{within}}(u, v) = 0.5(l)^{-2} \sum_{s,t=1}^{l} k(Y_{s,T}, Y_{t,T}) + 0.5(r-l)^{-2} \sum_{s,t=l+1}^{r} k(Y_{s,T}, Y_{t,T}),$$

where $l = \lceil vT \rceil$, $r = \lceil uT \rceil$, and the average similarity among observations between different groups is

$$\widehat{S}_T^{\text{between}}(u, v) = [l(r-l)]^{-1} \sum_{s=1}^{l} \sum_{t=l+1}^{r} k(Y_{s,T}, Y_{t,T}).$$

Intuitively, $k(y, y')$ should in general be larger if $y, y'$ follow the same distribution, and smaller if $y, y'$ follow different distributions. Thus, if $v$ is the true change point, we expect $\widehat{S}_T^{\text{within}}(u, v)$ to be large compared with $\widehat{S}_T^{\text{between}}(u, v)$. This intuition underlies the following statistic,

$$\widehat{\mathcal{D}}_T(u) = \max_{v \in [0,u]} \widehat{\mathcal{K}}_T(u, v) \quad \text{where} \tag{5}$$

$$\widehat{\mathcal{K}}_T(u, v) = 2v^2(u - v)^2/u^2 [\widehat{S}_T^{\text{within}}(u, v) - \widehat{S}_T^{\text{between}}(u, v)]. \tag{6}$$

$\widehat{\mathcal{D}}_T$ takes the maximum over $v \in [0, u]$ using a similar idea as $\widehat{\mathcal{D}}_T^{\text{uni}}$ and $\widehat{\mathcal{D}}_T^{\text{gen}}$. The scaling factor $v^2(u-v)^2/u^2$ is important and ensures that the limiting distribution of $\widehat{\mathcal{D}}_T$ is well-defined for all $u \in (0, 1]$ (see more details in Section 4). $\widehat{\mathcal{D}}_T$ plays the same role as $\widehat{\mathcal{D}}_T^{\text{gen}}$ and measures data variation among $[0, u]$.

Note that $\widehat{\mathcal{D}}_T$ has also a CUSUM-style representation, which is crucial for understanding its theoretical properties. Define a centered kernel $k_0(y, y') = k(y, y') - \mathbb{E}_{Y \sim P_0} k(y, Y) - \mathbb{E}_{Y \sim P_0} k(y', Y) + \mathbb{E}_{Y, Y' \sim P_0} k(Y, Y')$. Then $k_0$ can be decomposed in terms of eigenfunctions $\{\psi_j\}_{j=1}^{\infty}$ w.r.t. $P_0$ as:

$$k_0(y, y') = \sum_{j=1}^{\infty} \lambda_j \psi_j(y) \psi_j(y') \quad \text{with} \tag{7}$$

$$\int k_0(y, y') \psi_j(y) dP_0(y) = \lambda_j \psi_j(y'), \int \psi_j(y) \psi_{j'}(y) dP_0(y) = \delta_{j,j'},$$

and $\delta_{j,j'}$ is the Kronecker delta function. We denote the feature map $\phi$ associated with $k_0$ as

$$\phi(y) = (\lambda_1^{1/2} \psi_1(y), \lambda_2^{1/2} \psi_2(y), \cdots)^{\top} \in \mathcal{H}, \ \langle \phi(y), \phi(y') \rangle_{\mathcal{H}} := \sum_{l=1}^{\infty} \phi_l(y) \phi_l(y') = k_0(y, y').$$

Using properties of $\langle \cdot, \cdot \rangle_{\mathcal{H}}^{1/2}$ and denoting $\| \cdot \|_{\mathcal{H}} = \langle \cdot, \cdot \rangle_{\mathcal{H}}^{1/2}$, we have

$$\widehat{\mathcal{K}}_T(u, v) = \|1/T \sum_{t=1}^{[vT]} \phi(Y_{t,T}) - v/(uT) \sum_{t=1}^{[uT]} \phi(Y_{t,T})\|_{\mathcal{H}}^2 = \sum_{j=1}^{\infty} |\widehat{C}_T(u, v, \phi_j)|^2. \tag{8}$$

Equation (8) helps the comparison of $\widehat{\mathcal{D}}_T$ against $\widehat{\mathcal{D}}_T^{\text{gen}}$. In general, $\widehat{\mathcal{D}}_T$ has three advantages. First, recall that $\widehat{\mathcal{D}}_T^{\text{gen}}$ strongly depends on the specification of the function class $\mathcal{F}$; we allow implicitly a much larger $\mathcal{F}$ with infinite functions. For example, by using universal kernels such as $k(y, y') = \exp\{-\|y - y'\|_{\mathcal{Y}}^2/2\}$, we consider any change in $\mathbb{E}f(Y_{t,T}), f \in \mathcal{F}$ where $\mathcal{F}$ has infinite cardinality and satisfies the property that under mild assumptions, there always exists $f \in \mathcal{F}$ such that $\mathbb{E}f(X) \neq \mathbb{E}f(X')$ when random variables $X, X'$ follow different distributions. Second, the asymptotic distribution of $\widehat{\mathcal{D}}_T^{\text{gen}}$ (as $n$ grows to infinity) is intractable, caused by its dependence structure on $\widehat{C}_T$. There are two key facts, under $H_0$, for fixed $u, v$, as $T$ goes to infinity,

$$\widehat{C}_T(u, v, \phi_j) \xrightarrow{d} \text{Gaussian random variable, and } \mathbb{E}[\widehat{C}_T(u, v, \phi_j) \widehat{C}_T(u, v, \phi_{j'})] \to 0, \ \forall j, j' \in \mathbb{Z}_+.$$

It implies $\widehat{C}_T(u, v, \phi_j)$ are asymptotically independent Gaussian random variables (r.v.). Since the sum of squares of independent Gaussian r.v. follows a known distribution (chi-square), in view of (8), the asymptotic distribution of our statistic is much simpler than that of $\widehat{\mathcal{D}}_T^{\text{gen}}$. Third, using kernels to define $\widehat{\mathcal{K}}_T$ does not lead to technical/implementation issues. In contrast, if we define $\widehat{\mathcal{K}}_T$ directly using (8) with the function class $\mathcal{F} = \{\phi_j, j = 1, 2, \cdots\}$ replaced by an arbitrary function class of infinite cardinality, the infinite series will not necessarily converge, and even when it converges, it may not be calculated exactly. Using kernels, we circumvent this issue and with the trick mentioned in Appendix A, the total cost of calculating $\widehat{\mathcal{D}}_T(u)$ for all $u$'s takes $O(T^2)$ in both time and space.

*Remark* 3.1. Some useful kernels for the gradual CPD problem: (i) For $\mathcal{Y} = \mathbb{R}^d$, we recommend using the dot-product kernel $k(y, y') = \langle y, y' \rangle_{\mathbb{R}^d}$ if location model (2) holds. Here $\phi_j : x = (x_1, \cdots, x_d)^{\top} \mapsto x_j - \mathbb{E}_{P_0} X_j, \forall j = 1, \cdots, d$. When $d = 1$, $\widehat{\mathcal{D}}_T$ with this kernel equals $\widehat{\mathcal{D}}_T^{\text{gen}}$ with $\mathcal{F} = \{f : x \mapsto x - \mathbb{E}_{P_0} X\}$ and $\widehat{\mathcal{D}}_T^{\text{uni}}$. (ii) For $\mathcal{Y} = \mathbb{R}$, we recommend using $k(y, y') = y^2(y')^2$ if volatility model (3) holds. Here $\phi_j : x \mapsto x^2 - a$ where $a = \mathbb{E}_{X \sim P_0} X^2$. And $\widehat{\mathcal{D}}_T$ with this kernel equals $\widehat{\mathcal{D}}_T^{\text{gen}}$ with $\mathcal{F} = \{f : x \mapsto x^2 - a\}$. (iii) For any general $\mathcal{Y}$, $k(y, y') = \exp\{-\|y - y'\|_{\mathcal{Y}}^2/h\}$ is the RBF kernel with bandwidth $h > 0$. This can be set as the default kernel without any prior knowledge about data model.

Now we will utilize $\widehat{\mathcal{D}}_T$ for the detection and localization of gradual change points.

**Detection.** As shown in Figure 2, under a good choice of $k$, $\widehat{\mathcal{D}}_T(u)$ summarizes the degree of variation over time span $[0, u]$ and satisfies

$$\widehat{\mathcal{D}}_T(u) \text{ is } \begin{cases} \text{small}, & \text{when } u \leq \rho^*, \\ \text{large}, & \text{when } u > \rho^*. \end{cases} \tag{9}$$

The case of no change point is equivalent to $\rho^* = 1$. The existence of a change point can be tested using $\widehat{\mathcal{D}}_T(1)$. The p-value depends on the asymptotic null distribution of $\widehat{\mathcal{D}}_T(1)$, the rigorous establishment of which requires many technical details and is deferred to the next section (Theorem 4.4). Practitioners can use the following formula to calculate p-values:

$$\mathbb{P}(T\widehat{\mathcal{D}}_T(1) > x) \approx 2^{(\hat{q}+3)/2}[\Gamma(\hat{q}/2)]^{-1}\sqrt{\pi}(x/\hat{\lambda}_1)^{(\hat{q}-1)/2}e^{-2x/\hat{\lambda}_1}\prod_{l=q+1}^{T}(1 - \hat{\lambda}_l/\hat{\lambda}_1)^{-1/2}, \tag{10}$$

where $\hat{\lambda}_1 \geq \hat{\lambda}_2 \geq \cdots \geq \hat{\lambda}_T$ are eigenvalues of the matrix $(1/T)K_0$ where

$$K_0 = HKH \in \mathbb{R}^{T \times T}, \ K = [k(Y_{i,t}, Y_{j,T})]_{i,j=1}^{T} \in \mathbb{R}^{T \times T} \text{ and } H = I_T - (1/T)\mathbf{1}_T\mathbf{1}_T^{\top}, \tag{11}$$

and $\hat{q}$ is the estimated multiplicity of the leading eigenvalue. Accuracy of this approximation depends on the accuracy of estimated eigenvalues. In practice, we find it works well when $q$ is small (say, $q \leq 5$). When $q$ is large, we recommend estimating p-values by permutation tests.

**Localization.** Once a significant change point is detected, the next step is to localize it. Observing property (9) with $\widehat{\mathcal{D}}_T$ replaced by $\widehat{\mathcal{D}}_T^{\text{gen}}$, Vogt and Dette (2015) propose an estimator for $\rho^*$ as:

$$\hat{\rho}^{\text{gen}} = T^{-1}\sum_{t=1}^{T}\mathbb{I}(T^{1/2}\widehat{\mathcal{D}}_T^{\text{gen}}(t/T) \leq b_T^{\text{gen}}),$$

where the scaling factor $T^{1/2}$ ensures that $T^{1/2}\widehat{\mathcal{D}}_T^{\text{gen}}$ follows a non-degenerate distribution asymptotically as data size goes to infinity, and $b_T^{\text{gen}}$ is set to the $(1 - \alpha)$-quantile of the limiting distribution of $\sup_{v \in [0,\rho^*]} \widehat{\mathcal{D}}_T^{\text{gen}}(v)$. In practice, both $\rho^*$ and limiting distribution of $\widehat{\mathcal{D}}_T^{\text{gen}}(\cdot)$ are unknown, thus $b_T$ is approximated by a two-step procedure with expensive simulations. For our statistic, we find that under the null, $\widehat{\mathcal{D}}_T(u)$ and $u\widehat{\mathcal{D}}_T(1)$ follow the same limiting distribution for any $u$. It implies that we can estimate $\rho^*$ by

$$\hat{\rho} = T^{-1}\sum_{t=1}^{T}\mathbb{I}(T\widehat{\mathcal{D}}_T(t/T) \leq c_T(t/T)), \quad \text{where} \quad c_T(u) = ub_T, \tag{12}$$

and the scaling factor $T$ ensures that $T\widehat{\mathcal{D}}_T$ has a non-degenerate limiting distribution. Here, $\hat{\rho}$ is affected by $c_T$: a larger $c_T$ will lead to a larger $\hat{\rho}$ and vice versa. Ideally, the optimal choice of $c_T$ should minimize a measure of error, and we propose using $l_1(\hat{\rho}) = \mathbb{E}|\hat{\rho} - \rho^*|$. It depends on the finite sample distribution of $\widehat{\mathcal{D}}_T$ and could be hard to control in nonparametric settings, but we know the asymptotic distribution of $\widehat{\mathcal{D}}_T(\cdot)$ (Theorem 4.4). Thus, we choose the $c_T$ which minimizes the $l_1$ error of the population version $\rho^\infty$ of $\hat{\rho}$:

$$l_1(\rho^\infty) = \mathbb{E}|\rho^\infty - \rho^*| \quad \text{with} \quad \rho^\infty = \int_0^{\rho^*}\mathbb{I}(L_0(u) \leq c_T(u))du + \int_{\rho^*}^{1}\mathbb{I}(T^{1/2}L_1(u) + TD(u) \leq c_T(u))du,$$

where $L_0(\cdot), L_1(\cdot)$ correspond to the asymptotic distribution of properly re-scaled and re-centered $\widehat{\mathcal{D}}_T(\cdot)$ before and after $\rho^*$, respectively, and they are defined in Theorem 4.4. Under some assumptions, minimizing $l_1(\rho^\infty)$ leads to

$$b_T = \hat{\lambda}_1/(2\kappa)\log T, \tag{13}$$

where $\kappa \geq 2$ is determined by the smoothness of change and the smoother it is, the larger $\kappa$ is. The derivation of Equation (13) is included in the next section. The theoretical value of $\kappa$ is defined in Assumption 4, and it depends on the alternative distribution of $Y_{t,T}$ and the kernel $k$. For practitioners, we only need to know it for abrupt changes and any choice of kernel, $\kappa = 2$ (indeed, our method is also applicable for abrupt changes). For RBF, if the change in $\mathbb{E}\exp\{Y_{t,T}\}$ can be approximated by $(u - \rho^*)^\beta$ in time span $u \in [\rho^*, \rho^* + \varepsilon)$ for some small $\varepsilon > 0$, we have $\kappa = 2\beta + 2$. We emphasize that the choice of $\kappa$ does not affect the consistency of $\hat{\rho}$. In experiments, using rule of thumb $\kappa = 4$ works well. An alternative estimator that is less sensitive to $\kappa$ is introduced next.

**Max-gap estimator.** Despite its good theoretical properties, $\hat{\rho}$ has often a large positive bias. This arises from the nature of gradual changes, and is common to previous gradual CPD methods as discussed in Vogt and Dette (2015). Intuitively, we need to wait for enough signal strength in order to identify the gradual change point. To design a less biased estimator, recall that in Figure 2, we plotted $\widehat{\mathcal{D}}_T(\cdot)$ against time and easily visually identified the change point as the time when $\widehat{\mathcal{D}}_T(\cdot)$ starts to grow. For example, for data in the first column, a zoomed-in region is shown in Figure 3, where the black line is $T\widehat{\mathcal{D}}_T(\cdot)$ and red line $c_T(\cdot)$. In Figure 3, the growth starts around the point 285 (shown in

brown vertical line). However, using $\hat{\rho}$ gives $\hat{\tau} = 342$ (shown in green vertical line). We want an algorithm capable of identifying this elbow point (285). Note that from Theorem 4.4, we have

$$\mathbb{E}[c_T(u) - T\widehat{\mathcal{D}}_T(u)] \begin{cases} \text{increases with } u, & \text{if } u \leq \rho^* \\ \text{decreases with } u, & \text{if } u > \rho^*. \end{cases}$$

Thus, $\rho^*$ should be the $u$ where $c_T(u) - T\widehat{\mathcal{D}}_T(u)$ is maximized (in Figure 3, this is where the gap between the red line and black curve is maximized). It suggests setting

$$\check{\rho} = \mathrm{m\,arg\,max}_{u \in (0,\hat{\rho}]}[c_T(u) - T\widehat{\mathcal{D}}_T(u)], \qquad (14)$$

where $\mathrm{m\,arg\,max}$ takes the largest value in the set formed by $\arg\max$. In Figure 3, $\check{\rho}$ is shown by the brown line.

comparison of estimators

Figure 3: Comparison of max-gap estimator and original estimator in simulated data.

Compared with $\hat{\rho}$, empirical studies show two advantages of the max-gap estimator $\check{\rho}$: it is more accurate, and is much less sensitive to choice of $\kappa$. Some intuition for insensitivity to $\kappa$: in Figure 3, $\kappa$ changes the slope of the red line and a slight change in slope does not affect the time where its gap between the black line is maximized. The higher accuracy of $\check{\rho}$ also has a theoretical explanation, which is included in the Appendix due to space limit. In short, the $l_1$ error of $\check{\rho}$ consists of two parts: the overestimation error $\mathbb{E}[\check{\rho} - \rho^*]_+$ and the underestimation error $\mathbb{E}[\rho^* - \check{\rho}]_+$ with $[x]_+$ denotes the positive part of $x$. There is always a trade-off between overestimation and underestimation. Roughly, $\check{\rho}$ focuses more on controlling the overestimation error (delay) while guaranteeing consistency of the estimator, since delay is the main concern in small samples. In contrast, $\hat{\rho}$ controls the over/under-estimation error equally, which might be less accurate in small samples.

**Practical considerations.** All steps of the proposed procedure are summarized in Algorithm 1 in Section A of the Appendix. There we also discuss its time and space complexity.

## 4 Theory

This section establishes all theoretical results mentioned previously.

**Asymptotic distribution of $\widehat{\mathcal{D}}_T$.** In order to utilize $\widehat{\mathcal{D}}_T$ for downstream tasks, we need to know its asymptotic distribution. To establish that, we will first introduce some technical assumptions.

**Assumption 1.** $\exists M \in (0, +\infty)$, $\forall t \in \{1, 2, \cdots, T\}$, $k(Y_{t,T}, Y_{t,T}) \leq M^2$ almost surely (a.s.).

*Remark* 4.1. Assumption 1 requires that the kernel is a.s. bounded for all $Y_{t,T}$. It is a weak assumption which is satisfied when $k(\cdot, \cdot)$ is continuous and $\mathcal{Y}$ is closed and bounded, or when $k$ is RBF kernel.

Assumption 1 suffices for getting asymptotic null of $\widehat{\mathcal{D}}_T$. Under $H_A$, however, we need to restrict the changing pattern of $Y_{t,T}$: roughly, we require the change to be gradual, so the speed of change cannot be too fast compared with sample size. One useful concept to regulate such behavior is the locally stationary process, which has been used in Vogt and Dette (2015) for gradual CPD problems.

**Assumption 2** (Locally Stationary Process). *The array* $\{Y_{t,T} : t = 1, 2, \cdots, T\}_{T=1}^{\infty}$ *is a locally stationary process, i.e.,* $\forall u \in [0, 1]$, *there exists a strictly stationary process* $\{Y_t(u) : t \in \mathbb{Z}\}$ *s.t.*

$$\|Y_{t,T} - Y_t(u)\|_{\mathcal{Y}} \leq (|t/T - u| + 1/T)\, U_{t,T}(u) \quad a.s.$$

*where* $\{U_{t,T}(u) : t = 1, 2, \cdots, T\}_{T=1}^{\infty}$ *is an array of positive random variables which satisfies* $\mathbb{E}[U_{t,T}^{\gamma}(u)] \leq c_0$ *for some constant* $c_0 \in (0, +\infty)$, $\gamma > 0$.

*Remark* 4.2. Assumption 2 ensures that locally around each $u = t/T$, $\{Y_{t,T}\}$ can be approximated by a stationary process $\{Y_t(u)\}$. The constant $\gamma$ measures how well $Y_{t,T}$ is approximated by $Y_t(u)$: the larger $\gamma$ is, the better the approximation will be.

Define

$$\mathcal{D}(u) = \max_{v \in [0,u]} \mathcal{K}(u, v) \quad \text{with} \quad \mathcal{K}(u, v) = \|\textstyle\int_0^v \mu(w)dw - v/u \int_0^u \mu(w)dw\|_{\mathcal{H}}^2, \qquad (15)$$

where $\mu(\cdot) = (\mu_1(\cdot), \mu_2(\cdot), \cdots)^\top$, $\mu_j(\cdot) = \mathbb{E}\phi_j(Y_t(\cdot))$. Comparing Equations (15) and (8), we find that $\widehat{\mathcal{K}}_T(u, v)$ is in fact an estimator for $\mathcal{K}(u, v)$ and thus, $\widehat{\mathcal{D}}_T(u)$ is an estimator for $\mathcal{D}(u)$. Using the

decomposition $\widehat{\mathcal{D}}_T(u) = \mathcal{D}(u) + [\widehat{\mathcal{D}}_T(u) - \mathcal{D}(u)]$, in order to study asymptotics of $\widehat{\mathcal{D}}_T$, we only need to study the approximation error $\widehat{\mathcal{D}}_T - \mathcal{D}$. We will need the following assumptions:

**Assumption 3.** *The feature map $\phi$ and stochastic processes $\{\mu_j(u) : u \in [0,1]\}, \ \forall j \in \mathbb{Z}_+$ satisfy*

*(i) $\|\phi(y) - \phi(y')\|_{\mathcal{H}} \leq C_1 \|y - y'\|_{\mathcal{Y}}$ for all $y, y' \in \mathcal{Y}$.*

*(ii) $\gamma \geq 2$ where $\gamma$ is defined in Assumption 2.*

*(iii) $\sum_{j=1}^{\infty} \max_{u \in (0,1)} d\mu_j(u)/du < +\infty$.*

*Remark* 4.3. Condition (i) requires sufficient smoothness for $\phi$ which is always satisfied for sufficiently smooth kernels $k$. Intuitively, this helps us preserve the smoothness of the change in $Y_{t,T}$. Condition (ii) requires that $Y_{t,T}$ can be sufficiently well approximated by $Y_t(u)$ in the sense that $U_{t,T}(u)$ has finite variance. Condition (iii) roughly says that $\mu_j$ has a well-defined Riemann integral over $[0,1]$ so that the integral in $\mathcal{D}$ can be approximated by the Riemann sum in $\widehat{\mathcal{D}}_T$.

Now we are ready to present our main result, where $\rho^* = 1$ corresponds to no change point.

**Theorem 4.4.** *Suppose Assumption 1 holds.*

*(1) For any $u \in (0, \rho^*]$,*

$$T[\widehat{\mathcal{D}}_T(u) - \mathcal{D}(u)] \xrightarrow{w} \max_{v \in [0,u]} \sum_{l=1}^{\infty} \lambda_l [W_l(v) - \tfrac{v}{u} W_l(u)]^2 =: L_0(u), \qquad (16)$$

*where $\lambda_l$'s are defined in (7), and $W_l(\cdot), l = 1, \cdots$ are independent standard Wiener processes.*

*(2) If, in addition, Assumptions 2 and 3 hold, for any $u \in (\rho^*, 1]$, we have*

$$\sqrt{T}[\widehat{\mathcal{D}}_T(u) - \mathcal{D}(u)] \xrightarrow{w} \max_{v \in [0,u]} G(v, u) =: L_1(u), \qquad (17)$$

*where for any $u$, $G(\cdot, u)$ is a sample continuous Gaussian process.*

*Remark* 4.5. Both $\sum_{l=1}^{\infty} \lambda_l [W_l(\cdot) - \tfrac{\cdot}{u} W_l(u)]^2$ and $G(\cdot, u)$ are sample continuous and thus, the right hand size of (16) (17) are well-defined. $\lambda_l$'s are determined by $P_0$, $k$ and (16) states that the higher the noise level of $P_0$ is, the more dispersed the asymptotic null of $\widehat{\mathcal{D}}_T$ will be. Note that the asymptotic distribution of $\widehat{\mathcal{D}}_T$ is quite different before and after the change point: before change point, for each $u$, $\widehat{\mathcal{D}}_T(u) = O_p(T^{-1})$ and after re-scaling, $\widehat{\mathcal{D}}_T(u)$ is maximum of a chi-square process; after change point, $\widehat{\mathcal{D}}_T(u) = \mathcal{D}(u) + O_p(T^{-1/2})$ and after re-centering and re-scaling, $\widehat{\mathcal{D}}_T(u)$ is maximum of a Gaussian process. This distinct property of $\widehat{\mathcal{D}}_T(\cdot)$ is critical for the success of the proposed procedure.

**Detection.** To calculate p-values, Theorem 2.1 of Liu and Ji (2014) says that for $\forall n \in \mathbb{Z}_+$ and $\lambda_1 = \cdots = \lambda_q > \lambda_{q+1} \geq \lambda_{q+2} \geq \cdots \geq \lambda_n > 0$, as $x \to \infty$,

$$\mathbb{P}(\max_{v \in [0,1]} \sum_{l=1}^{n} \lambda_l [W_l(v) - v W_l(1)]^2 > x)$$
$$= 2^{(q+3)/2} [\Gamma(q/2)]^{-1} \sqrt{\pi} (x/\lambda_1)^{(q-1)/2} \exp\{-2x/\lambda_1\} \prod_{l=q+1}^{n} (1 - \lambda_l/\lambda_1)^{-1/2} (1 + o(1)).$$

Combined with Theorem 4.4, it implies (10). Also, we have the following:

**Corollary 4.1** (Power Consistency)**.** *Suppose Assumption 1, 2, 3 hold. If $\sqrt{T}\mathcal{D}(1) \to \infty$,*

$$\forall x > 0, \quad \mathbb{P}(T\widehat{\mathcal{D}}_T(1) > x) \to 1, \quad T \to \infty.$$

*Remark* 4.6. Corollary 4.1 shows that power of the proposed test is affected by the magnitude of change measured in $\mathcal{D}(1)$. As long as $\mathcal{D}(1)$ goes to zero at a rate slower than $T^{-1/2}$, the change will be detected if it exists; it ensures correctness of the detection step.

**Localization.** Recall we need to optimize $c_T$. This requires regulating the local behavior of $\mathcal{D}$ at $\rho^*$:

**Assumption 4.** *There is a cusp of order $\kappa$ at $\rho^*$ for $\mathcal{D}(\cdot)$, i.e., $\frac{\mathcal{D}(u)}{(u-\rho^*)^\kappa} \to m > 0, \quad u \to \rho^* +$ .*

*Remark* 4.7. Assumption 4 says $\mathcal{D}$ can be locally approximated by a Taylor-type expansion around $\rho^*$, which is a common assumption for gradual CPD (Mallik et al., 2013; Vogt and Dette, 2015).

**Theorem 4.8.** *Suppose Assumptions 1, 2, 3, 4 hold, and $c_T(u) = ub_T$. The $c_T$ minimizing $l_1(\rho^\infty)$ satisfies*

$$c_T(u) = (u\lambda_1 r \log T)/2, \ r \geq 1/\kappa. \qquad (18)$$

*Remark* 4.9. In Equation (18), the larger the noise level $\lambda_1$ is, the larger $c_T$ is. The smoother the gradual change is (the larger $\kappa$ is), the smaller $c_T$ is. And $r$ can be viewed as a tuning parameter s.t. if we are less tolerant to delays in $\hat{\rho}$, we could set $r$ to be small, and vice versa. In practice, $\hat{\rho}$ is often overestimated. Thus, we suggest choosing $r = 1/\kappa$, which ultimately leads to (13).

**Theorem 4.10.** *Under Assumptions 1, 2, 3, 4 and Equation* (18), $\hat{\rho} - \rho^* = o_p(1)$, $\check{\rho} - \rho^* = o_p(1)$.

*Remark* 4.11. Theorem 4.10 shows that the original estimator and the max-gap estimator are both consistent, and establishes theoretical guarantees for the localization step.

# 5  Simulations

To better understand finite sample properties of the proposed method, we evaluate its performance in simulations and against baselines. Additional details and results including type I error (p-value calibration), power comparison and performance comparison on strings are included in the Appendix.

**Data generating process.** We set $\rho^* = 1/3$. Following Vogt and Dette (2015), we consider a location model, a volatility model, and we add a network model. For the location model (2), we include univariate cases with $\varepsilon_t \sim N(0,1)$ and four different types of change ordered in increasing difficulty: (i) linear change $\mu_1(u) = \mathbb{I}(1/3 \leq u \leq 2/3)(3u-1) + \mathbb{I}(u \geq 2/3)$; (ii) quadratic change $\mu_2(u) = \mathbb{I}(1/3 \leq u \leq 2/3)(3u-1)^2 + \mathbb{I}(u \geq 2/3)$; (iii) one-sided change $\mu_3(u) = 2\sin(2.5\pi(u-1/3))\mathbb{I}(1/3 \leq u \leq 2/3) + \mathbb{I}(u \geq 2/3)$ in the sense that $\mu_3(u) > \mu_3(\rho^*)$ for all $u > \rho^*$; and (iv) a complex change $\mu_4(u) = 2\sin(4\pi(u-1/3))\mathbb{I}(1/3 \leq u \leq 2/3) + 2\sin(4\pi/3)\mathbb{I}(u \geq 2/3)$. We also consider multivariate $Y_{t,T} \in \mathbb{R}^d$ where $\mu_5 = \mu_1 \mathbf{1}_d, \varepsilon_t \sim N_d(0, I_d)$. For volatility model (3), we consider $\sigma_i(u) = \mu_i(u) + 1, \varepsilon_t \sim N(0,1), \forall i = 1, 4$. For network model, we set $Y_{t,T}$ as the Erdos-Renyi random graph with 10 nodes. At each time $u \in [0,1]$, there exists a 3-node community such that the possibility of forming an edge among them follows Binomial$(1, p(u))$ independently. Here $p(u) = 0.8\mathbb{I}(1/3 \leq u \leq 2/3)(3u-1) + 0.8\mathbb{I}(u \geq 2/3) + 0.1$. The probability of forming an edge between other pair of nodes always follows a Binomial$(1, 0.1)$.

**Baselines.** We consider four gradual CPD baselines, ordered in increasing generality: $\hat{\rho}^{\text{poly}}$ (Hušková, 1999) which requires univariate location model with polynomial change, $\hat{\rho}^{\text{one-side}}$ (Mallik et al., 2013) which requires univariate location model with one-sided change, $\hat{\rho}^{\text{mix}}$ (Quessy, 2019) which requires any general model with a mixture type of change whose mixture weight changes linearly with time, and $\hat{\rho}^{\text{gen}}$ (Vogt and Dette, 2015) which does not have any particular constraints on model or type of change. We also include three nonparametric abrupt CPD methods: KCpA (Harchaoui et al., 2008), $Z_w$ (Chu et al., 2019), and $Q$ (Matteson and James, 2014)).

**Detailed setup.** Setting I (main experiment): We set $T = 600$. For $\hat{\rho}^{\text{one-side}}$ we tune the bandwidth on 20 independently generated datasets among $\{0.01, 0.05, 0.1, 0.2, 0.3, 0.4, 0.5, 0.6, 0.7, 0.8, 0.9, 1, 5\}$. For each dataset, for fairness we use the same kernel for $\hat{\rho}, \check{\rho}$ and KCpA, and use its corresponding distance for $Q, Z_w$ and function class $\mathcal{F}$ for $\hat{\rho}^{\text{gen}}$. For location model, $\mathcal{F} = \{f : x \mapsto x_i, \forall i = 1, \cdots, d\}$; for network model, $\mathcal{F} = \{f : x \mapsto x_{ij}, \forall i, j = 1, \cdots, 10\}$; for volatility model, $\mathcal{F} = \{f : x \mapsto x^2\}$. For $\hat{\rho}^{\text{poly}}$ we set the polynomial to the true degree if the polynomial model is correct, and 1 otherwise. As recommended by their authors, we use a granularity of 20 for $\hat{\rho}^{\text{mix}}$ and minimum spanning tree to construct the binary graph for $Z_w$. Threshold for $\hat{\rho}^{\text{gen}}$ is computed using strategy described in Section 6 of Vogt and Dette (2015).

Setting II (influence of bandwidth): We note that both the proposed method and KCpA are kernel-based. In setting I, we use the kernel that is theoretically best for both of them. As suggested by reviewers, in this setting, we search for the empirically best RBF kernel $k(y, y') = \exp\{-\|y - y'\|^2/h\}$ where $h$ is the bandwidth and is tuned among $\{0.01, 0.05, 0.1, 1, 5, 10, 20, 50, 100, 500\}$ on 20 independently generated data sets. Here $\|\cdot\|$ is the $l_2$ distance for scalars/vectors and Frobenius norm for network. We set $T = 210$ and report the testing performance on 20 separate testing sets.

**Metrics and Results.** We report the power and $l_1$ error of estimated change points. For fairness, power of all methods are computed via 500 permutations under significance level $\alpha = 0.05$. Due to space limit, detailed results on power are included in the Appendix - performance of all abrupt as well as gradual CPD methods are similar. In terms of localization, however, performance varies. In Table 1a, the abrupt CPD methods (KCpA, $Q$, $Z_w$) have a large error in most settings, which is not surprising because KCpA, $Q$ are designed for abrupt changes. For $\hat{\rho}^{\text{poly}}, \hat{\rho}^{\text{one-side}}$ which require assumptions on the changing form, the localization is accurate when assumptions are satisfied, but

Table 1: Comparison of average $l_1$ localization error over 20 simulations. Numbers after $\pm$ are the standard error of the average. Methods marked with '-' means not applicable to that model.

(a) Setting I.

| MODEL | LOCATION | | | | | | | VOLATILITY | | NETWORK |
|---|---|---|---|---|---|---|---|---|---|---|
| DIM | 1 | 1 | 1 | 1 | 10 | 20 | 50 | 1 | 1 | $10^2$ |
| CHANGE | LINEAR | QUADRATIC | ONE-SIDED | COMPLEX | LINEAR | LINEAR | LINEAR | LINEAR | COMPLEX | LINEAR |
| $\check{\rho}$ | 0.09±0.01 | 0.15±0.01 | 0.03±0.00 | **0.03±0.01** | 0.07±0.01 | **0.06±0.01** | 0.05±0.01 | 0.15±0.01 | **0.05±0.00** | **0.10±0.02** |
| $\hat{\rho}$ | 0.10±0.01 | 0.24±0.01 | 0.08±0.00 | 0.05±0.01 | 0.08±0.01 | 0.10±0.01 | 0.09±0.01 | 0.26±0.01 | 0.12±0.00 | 0.11±0.02 |
| $\hat{\rho}^{\mathrm{POLY}}$ | **0.05±0.01** | **0.09±0.02** | 0.10±0.01 | 0.23±0.00 | - | - | - | - | - | - |
| $\hat{\rho}^{\mathrm{ONE\text{-}SIDE}}$ | 0.07±0.01 | **0.09±0.02** | **0.02±0.00** | 0.62±0.00 | - | - | - | - | - | - |
| $\hat{\rho}^{\mathrm{MIX}}$ | **0.05±0.01** | **0.09±0.01** | 0.14±0.00 | 0.12±0.00 | 0.08±0.00 | 0.18±0.02 | 0.43±0.00 | **0.08±0.01** | 0.14±0.00 | - |
| $\hat{\rho}^{\mathrm{GEN}}$ | 0.17±0.01 | 0.24±0.01 | 0.07±0.00 | 0.05±0.00 | 0.13±0.01 | 0.15±0.01 | 0.14±0.00 | 0.26±0.01 | 0.12±0.00 | 0.27±0.00 |
| $Q$ | 0.18±0.01 | 0.23±0.01 | 0.05±0.00 | 0.27±0.00 | 0.16±0.01 | 0.17±0.00 | 0.16±0.00 | 0.21±0.01 | 0.06±0.00 | 0.16±0.01 |
| KCPA | 0.18±0.01 | 0.23±0.01 | 0.05±0.00 | 0.27±0.00 | 0.16±0.01 | 0.16±0.00 | 0.16±0.00 | 0.21±0.01 | 0.06±0.00 | 0.16±0.01 |
| $Z_w$ | 0.24±0.04 | 0.29±0.04 | 0.09±0.02 | 0.29±0.01 | 0.16±0.01 | 0.17±0.01 | 0.18±0.01 | 0.18±0.03 | 0.16±0.03 | 0.16±0.02 |

(b) Setting II.

| MODEL | LOCATION | | | | | | | VOLATILITY | | NETWORK |
|---|---|---|---|---|---|---|---|---|---|---|
| DIM | 1 | 1 | 1 | 1 | 10 | 20 | 50 | 1 | 1 | $10^2$ |
| CHANGE | LINEAR | QUADRATIC | ONE-SIDED | COMPLEX | LINEAR | LINEAR | LINEAR | LINEAR | COMPLEX | LINEAR |
| $\check{\rho}$ | 0.14±0.01 | 0.19±0.01 | **0.05±0.01** | **0.04±0.01** | 0.07±0.01 | 0.05±0.01 | 0.08±0.01 | 0.15±0.01 | 0.13±0.01 | 0.02±0.01 |
| $\hat{\rho}$ | **0.12±0.01** | **0.18±0.01** | 0.09±0.01 | 0.07±0.01 | 0.09±0.01 | 0.07±0.01 | 0.11±0.01 | **0.15±0.01** | 0.21±0.01 | 0.07±0.01 |
| KCPA | 0.18±0.01 | 0.23±0.01 | **0.05±0.01** | 0.29±0.01 | 0.21±0.01 | 0.16±0.01 | 0.18±0.01 | 0.28±0.02 | 0.23±0.06 | 0.17±0.01 |

poor otherwise. $\hat{\rho}^{\mathrm{mix}}$ performs well in low dimensions and when the change (approximately) satisfies its assumption, but poorly when either one is violated. The proposed estimators $\hat{\rho}, \check{\rho}$ are robust across different settings and $\check{\rho}$ has improved performance over $\hat{\rho}$. $\hat{\rho}^{\mathrm{gen}}$ is also significantly outperformed by $\check{\rho}$. Finally, note $\hat{\rho}^{\mathrm{mix}}, \hat{\rho}^{\mathrm{gen}}$ are more computationally expensive than the others. In Table 1b, the conclusion is consistent where we use RBF kernel with tuned bandwidth. Together, Table 1 shows the advantage of $\hat{\rho}, \check{\rho}$ in terms of handling general types of data and general types of changes.

## 6 Real Data Applications

Different from most machine learning tasks, there are currently no benchmarking dataset with human annotations for gradual CPD. Thus, we consider the applications introduced in Section 1, and compare our result with known external events and/or other CPD estimators.

**Central England Temperature.** The Central England Temperature (CET) record (Parker et al., 1992) under Open Government License is the oldest temperature record worldwide and is a valuable source for studying climate change. It contains the monthly mean temperature in central England from 1750 to 2020. Since there is a cycle of 12 months for the measurements, following Horváth et al. (1999), we view the data as $n = 271$ curves with 12 measurements on each curve. We set $k(y, y') = y^\top y'$ where $y, y' \in \mathbb{R}^{12}$. Using max-gap estimator, we identify 1827 as the change point (shown in red vertical line in Figure 1a), which roughly corresponds to the beginning of mass industrialization and is close to the 1850 estimated by Berkes et al. (2009).

**S&P 500 Index.** The S&P 500 is a stock market index which tracks the stock of 500 large US companies and is usually used as a benchmark of the overall market. We investigate the daily return data of the S&P 500 index[1] in two periods, one from 2008/01/02 to 2008/12/31 and another from 2019/06/03 to 2020/06/01. Both time periods contain a change point where volatility level gradually increases. Following Vogt and Dette (2015), the daily return $Y_{t,T}$ roughly follows the volatility model (3) and our task is to identify changes in $\sigma(\cdot)$. We define the kernel as $k(y, y') = y^2(y')^2$ where $y, y' \in \mathbb{R}$. In both periods, we detect a change under $\alpha = 0.05$. The first period has an estimated change point 2008/09/16, following Lehman Brothers Bankruptcy in September 15 which is often viewed as a turning point in the crisis. The second period has an estimated change point 2020/02/24, days in the initial phase of the community spread of COVID-19 in the United States. The estimated change points are shown in red vertical lines in Figure 1b.

## 7 Related Work

Here we discuss some related work, with some additional reviews included in Section E in Appendix.

---

[1]S&P Dow Jones Indices LLC, S&P 500 [SP500], retrieved from https://finance.yahoo.com/quote/%5EGSPC/history/.

**Difference with Vogt and Dette (2015).** The major improvements of this work over Vogt and Dette (2015) are discussed in detail in Sections 1, 3. Other differences include: Vogt and Dette (2015) allow correlated observations, while we assume independence; Vogt and Dette (2015) uses estimator (3), while we propose a refined max-gap estimator that performs better empirically. We note that the our method might also be adapted for the correlated case, a possible direction for future work.

**Abrupt CPD.** Abrupt CPD methods assume the distribution remains stationary until the change point when it jumps to another distribution, and remains stationary there. There is a rich literature on them; see Niu et al. (2016); Aminikhanghahi and Cook (2017); Truong et al. (2020) for detailed surveys. In our experiments, we find that abrupt CPD methods seem to produce poor localization estimators for gradual changes. Here we explain this phenomenon by giving a toy example in which the abrupt CPD methods fail to be consistent. Consider the case where $Y_{i,T} = \mu_1(i/T)$ with $\mu_1(u) = \mathbb{I}(1/3 \leq u \leq 2/3)(3u - 1) + \mathbb{I}(u \geq 2/3)$. Recall that all three abrupt CPD methods we consider (KCpA, $Q$, $Z_w$) estimate the change point as

$$\tilde{\rho} := \arg\max_t d(P_0(t), P_1(t)), \tag{19}$$

where $d(P_0(t), P_1(t))$ is some (standardized) discrepancy measure between the two groups of data separated by $t$. Notice that Equation (19) differs fundamentally from the proposed estimator (12). For $Q$, if we set $\alpha = 2$ in Equation (4) of Matteson and James (2014), one can easily show that $\tilde{\rho}$ converges to $1/2$, which is different from the truth $1/3$. This same observation also holds for KCpA and $Z_w$ with more complicated analysis, but the general observation is that estimators of the form (19) may fail to be consistent under some gradual changes.

**MMD.** The Maximum Mean Discrepancy (MMD) is proposed in Gretton et al. (2012) for two-sample tests. We note that our intermediate statistic $\widehat{\mathcal{K}}_T(u, v)$ is similar to MMD. However, we emphasize the differences here: First, MMD is designed for two-sample tests where all theoretical analyses are based on two fixed samples, while we aim at the change point problem where we do not know where the true change point is. Thus, our final statistic $\widehat{\mathcal{D}}_T$ requires extra and careful handling (e.g., the scaling factor $v^2(u - v)^2/u^2$ and the max over $v \in [0, u]$). These extra terms complicate the theoretical analysis (see Theorem 4.4). Second, the alternative in two-sample tests is that the two groups of data follow two different distributions; however, with gradual change point problems, we really do not have two distinct distributions but an infinite number of different distributions! Indeed, the main theoretical difficulty of this work lies in characterizing the behavior of $\widehat{\mathcal{D}}_T$ under the alternative with gradual changes, which is quite different from the (local) alternatives in two-sample tests and requires more involved analysis. Further, the more difficult task we address is to localize the change point (instead of detecting change points), which does not exist for two sample tests.

**CUSUM.** The CUSUM principle was proposed by Page (1954) and has led to a rich literature. Some papers have investigated using CUSUM under gradual changes (Bissell, 1984a,b; Gan, 1992), but they considered only simple settings with a linear trend in the mean of univariate data, and their analyses are based mostly on empirical studies.

**Kernel-based CPD methods.** Existing kernel-based CPD methods all focus on the abrupt settings (Harchaoui et al., 2008; Arlot et al., 2012; Li et al., 2015; Garreau et al., 2018). We emphasize that their method is fundamentally different from ours, and, as far as we know, none of them produces a consistent localization estimator in the settings considered in this paper.

## 8 Discussion

We propose a general method to detect and to localize gradual changes in sequence data. Despite the relaxed assumptions, the proposed method is theoretically guaranteed, and the proposed max-gap estimator achieves good empirical performance. Note that the proposed method also works for abrupt CPD with Corollary 4.1 and Theorem 4.10 hold. In contrast, many abrupt CPD methods perform poorly in gradual change settings. The trade-off is that for abrupt changes or gradual changes with a known pattern (e.g., polynomial), our method often does not perform as good as those designed especially for those settings. There are no foreseeable negative social impacts of this work.

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
