# A  Summary of the Proposed Procedure

We summarize all steps required to detect and to localize change point in the following Algorithm 1.

---

**Algorithm 1** Gradual Change Point Detection and Localization

---

1: **input**: Kernel matrix $K$ of data $\{Y_{t,T}\}$, significance level $\alpha$, smoothness of change point $\kappa$, the number of nonzero eigenvalues $n \leq T$.
2: **output**: estimated change point location $\hat{\rho}$.  $\triangleright$ $\hat{\rho} = 1$ implies no significant change point
3: **prepare**:
4:  compute $\widehat{\mathcal{D}}_T(t/T)$ for $t = 1, 2, \cdots, T$ using Equation (5).
5:  compute the first $n$ eigenvalues $\hat{\lambda}_1, \cdots, \hat{\lambda}_n$ of $(1/T)K_0$ using Equation (5) and (11).
6: **detection**:
7:  obtain p-value of $\widehat{\mathcal{D}}_T(1)$ using Equation (10) or permutation tests.
8:  **if** p-value $> \alpha$:  **return** $\hat{\rho} = 1$  $\triangleright$ accept the null
9:  **else**:  go to line 9.  $\triangleright$ reject the null
10: **localization:**
11:  calculate $b_T$ using Equation (13).
12:  estimate $\hat{\rho}$ using Equation (12).
13:  (optional) obtain refined estimator using (14).  $\triangleright$ max-gap estimator
14:  **return** $\hat{\rho}$ (or $\check{\rho}$ if it is computed).

---

Note that in line 5 of Algorithm 1, we compute the first $n \leq T$ eigenvalues for $K_0$, where $n$ is chosen a prior if one believes that the kernel contains at most $n$ nonzero eigenvalues. This effectively reduces the computational cost: for example, if one believes $n = 1$, power iteration can be applied to obtain $\hat{\lambda}_1$ which takes only $O(T^2)$ time and is much cheaper than obtaining the whole spectrum.

**Computational complexity analysis.**  Depending on how we calculate p-value, the proposed procedure shown in Algorithm 1 takes $O(BT^2)$ time complexity with $B$ number of permutations if we use permutation tests or $O(nT^2)$ if we use Equation (10). Detailed explanation: one can pre-compute a $T \times T$ matrix where the element on $i$-th row and $j$-th column denotes the cumulative sums $\sum_{l=1}^{i} \sum_{m=1}^{j} k(Y_{l,T}, Y_{m,T})$. Then each $\widehat{\mathcal{D}}_T(t/T)$ can be computed with an additional $O(T)$ time. The detection procedure, if using approximation (10), takes $O(T^3)$ for obtaining all eigenvalues; in practice, however, we may only need to compute the first $n \ll T$ eigenvalues which takes $O(nT^2)$; if the p-value is computed using permutation tests, it takes $O(BT^2)$ where $B$ is the number of permutations we use. The localization procedure takes $O(T^2)$ since we only need the largest eigenvalue $\lambda_1$ and it can be computed using power iteration with $O(T^2)$ complexity.

# B  Further Discussions on Improving Localization Accuracy

This section gives some further discussions on (a) why the max-gap estimator has improved accuracy over the original estimator, and (b) what are other ways to improve accuracy of the original estimator.

**Notations.**  Denote $[x]_+ = \max(x, 0)$. For any estimator $\hat{\rho}$, define the overestimation error as $\mathbb{E}[\hat{\rho} - \rho^*]_+$, the underestimation error as $\mathbb{E}[\rho^* - \hat{\rho}]_+$, and the overall error as $\mathbb{E}|\rho^* - \hat{\rho}|$.

## B.1  Comparison of Original Estimator and the Max-Gap Estimator

As discussed in the main paper, the higher accuracy of the max-gap estimator compared with the original estimator has a theoretical explanation. Define function $w : [0, 1] \to \mathbb{R}$ such that $w(u) = 1$ for $u \in [0, \rho^*]$, $w(u) = T^{1/2}$ for $u \in (\rho^*, 1]$. Theorem 4.4 says that

$$T[w(u)]^{-1}(\widehat{\mathcal{D}}_T(u) - \mathcal{D}(u)) \xrightarrow{w} L(u),$$

where $L(u) = L_0(u)$ if $u \in (0, \rho^*)$ and $L(u) = L_1(u)$ if $u \in (\rho^*, 1]$. We note that $L(u)$ is a well-defined, non-degenerate distribution. Thus, $T\mathcal{D}(u) + w(u)L(u)$ can be viewed as the population version of $T\widehat{\mathcal{D}}_T(u)$. Define the population version of $\check{\rho}$ as $\rho_\infty$:

$$\rho_\infty = \arg\max_{u \in [0,1]} \left[ c_T(u) - (T\mathcal{D}(u) + w(u)L(u)) \right].$$

We have the following result:

**Theorem B.1.** *Under Assumptions 1, 2, 3, 4, if $b_T \to \infty$, $b_T/T \to 0$ as $T \to +\infty$, we have*

*original estimator:* $[\rho^\infty - \rho^*]_+ = O_p\left(\left(\frac{b_T}{T}\right)^{1/\kappa}\right)$, $[\rho^* - \rho^\infty]_+ = O_p\left(P(b_T)\exp\left(-\frac{2b_T}{\lambda_1}\right)\right)$,

*max-gap estimator:* $[\rho_\infty - \rho^*]_+ = o_p\left(\left(\frac{b_T}{T}\right)^{1/\kappa}\right)$, $[\rho^* - \rho_\infty]_+ = O_p\left(\frac{1}{b_T}\right)$.

*where $P(b_T)$ is a known function having polynomially bounded growth as $b_T \to \infty$.*

*Remark* B.2. Theorem B.1 shows that when $b_T = \Theta(\log T)$, in the population sense, the max-gap estimator control of the overestimation/underestimation error has different strengths: compared with the original estimator, it has a smaller overestimation error and larger underestimation error. Since overestimation is the major concern in small sample size, the max-gap estimator has usually higher overall accuracy than the original estimator in experiments.

The above analysis is based on the population versions of the estimators. We also conducted some experiments to see whether the conclusion holds for finite samples. In Figure 4, we see that the max-gap estimator performs much better than the original one in terms of overestimation, and slightly worse in terms of underestimation, which in turn, results in a smaller overall error.

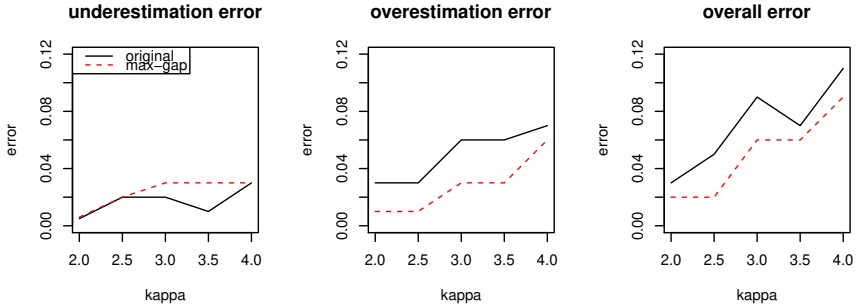

Figure 4: Empirical comparison of $\hat{\rho}$ and $\check{\rho}$ in simulated dataset with $P_u = N(\mu(u), 0.5^2)$, $\mu(u) = (3u-1)^{\kappa/2-1}\mathbb{I}(1/3 \le u \le 2/3) + \mathbb{I}(u > 2/3)$. We set $T = 600$, and errors are averaged over 10 simulations.

## B.2 Bias Correction of Original Estimator

Under some special cases, there is another way to improve the performance of the original estimator $\hat{\rho}$. Recall that $\rho^\infty$ is the population version of $\hat{\rho}$. First, let us introduce a general result:

**Theorem B.3.** *Under Assumptions 1, 2, 3, 4, if $b_T$ satisfies $b_T > \lambda_1 \log T/(2\kappa)$ and $b_T/T \to 0$ as $T \to +\infty$, when $T$ is sufficiently large, we have*

$$\left(\frac{\rho^* b_T}{mT}\right)^{1/\kappa} + o((\log T/T)^{1/\kappa}) \le \mathbb{E}\left[\rho^\infty - \rho^*\right]_+$$

$$\le \left(\frac{b_T}{mT}\left[\sqrt{\rho^* + \sigma_{\max}^2/\lambda_1} + \sqrt{\sigma_{\max}^2/\lambda_1}\right]^2\right)^{1/\kappa} + o((\log T/T)^{1/\kappa}), \quad (20)$$

*where $\sigma_{\max}^2 = \sup_{l,u} Var\left(\phi_l(Y_t(u)\right)$ and $m, \kappa$ are defined in Assumption 4. Notice that we have $\sigma_{\max}^2 \ge \lambda_1$.*

*Remark* B.4. Note that the left hand side and right hand size in Equation (20) are of the same order, which implies $\mathbb{E}\left[\rho^\infty - \rho^*\right]_+ = \Theta\left(\left(\frac{b_T}{mT}\right)^{1/\kappa}\right)$. Thus, the delay in $\rho^\infty$ roughly increases with larger threshold $b_T$, smaller sample size $T$ and smoother change point (larger $\kappa$).

In order to alleviate the large delay in $\hat{\rho}$, we can subtract the lower bound in Theorem B.3 from the estimated location $\hat{\rho}$. This requires knowledge about $\rho^*, m, \kappa$, the latter two of which are hard to estimate in general. However, under some particular settings, we can estimate $m, \kappa$ using empirical samples. One such simplified setting is when there is a stationary distribution at the beginning and the end of the sequence, and the changing phase in the middle is a mixture of $P_0, P_1$:

$$
P_u = \begin{cases} P_0, & u \in [0, \rho_0], \\ \frac{\rho_1 - u}{\rho_1 - \rho_0} P_0 + \frac{u - \rho_0}{\rho_1 - \rho_0} P_1, & u \in (\rho_0, \rho_1], \\ P_1, & u \in (\rho_1, 1], \end{cases} \tag{21}
$$

where $\rho_0 = \rho^*$, and $\rho_1 \in (\rho^*, 1)$. Thus $\hat{\rho}$ is essentially an estimator for $\rho_0$, so we also denote it as $\hat{\rho}_0$. The estimator for $\rho_1$ can be obtained using exactly the same way as that for $\hat{\rho}_0$ by reversing the sequence.

**Corollary B.1.** *Suppose all assumptions in Theorem B.3 hold.*

*(1) If $P_u$ satisfies Equation* (21), *when $T$ is sufficiently large, we have the following bound:*

$$
\left( \frac{2(\rho_1 - \rho_0)}{\|\mu_0 - \mu_1\|_{\mathcal{H}}} \sqrt{\rho_0} \right)^{1/2} \left( \frac{b_T}{T} \right)^{1/4} + o((\log T/T)^{1/4}) \leq \mathbb{E}\left[ \rho^\infty - \rho_0 \right]_+
$$

$$
\leq \left[ \frac{2(\rho_1 - \rho_0)}{\|\mu_0 - \mu_1\|_{\mathcal{H}}} \left( \sqrt{\rho_0 + \sigma_{\max}^2/\lambda_1} + \sqrt{\sigma_{\max}^2/\lambda_1} \right) \right]^{1/2} \left( \frac{b_T}{T} \right)^{1/4} + o((\log T/T)^{1/4}),
$$

*where $\mu_0 = \mathbb{E}_{P_0} \phi(Y)$, $\mu_1 = \mathbb{E}_{P_1} \phi(Y)$.*

*(2) If changes are abrupt, i.e., $\rho_0 = \rho_1$, we have*

$$
\frac{\sqrt{2\rho_0}}{\|\mu_0 - \mu_1\|_{\mathcal{H}}} \left( \frac{b_T}{T} \right)^{1/2} \leq \mathbb{E}\left[ \rho^\infty - \rho_0 \right]_+ \leq \frac{\sqrt{2\rho_0 + 2\sigma_{\max}^2/\lambda_1} + \sqrt{2\sigma_{\max}^2/\lambda_1}}{\|\mu_0 - \mu_1\|_{\mathcal{H}}} \left( \frac{b_T}{T} \right)^{1/2}.
$$

In practice, we do not know the truth $\|\mu_0 - \mu_1\|_{\mathcal{H}}$. But once we get an initial estimator $(\hat{\rho}_0, \hat{\rho}_1)$, we can estimate it using the following formula:

$$
\left[ \frac{1}{n_0(n - n_1)} \sum_{i=1}^{n_0} \sum_{j=n_1}^{n} k(Y_{i,T}, Y_{j,T}) - \frac{1}{2n_0^2} \sum_{i,j=1}^{n_0} k(Y_{i,T}, Y_{j,T}) - \frac{1}{2(n - n_1)^2} \sum_{i,j=n_1+1}^{n} k(Y_{i,T}, Y_{j,T}) \right]^{1/2}. \tag{22}
$$

with $n_0 = \lceil T\hat{\rho}_0 \rceil, n_1 = \lceil T\hat{\rho}_1 \rceil$.

We note that in the special setting of (21), the performance of the bias corrected estimator

$$
\hat{\rho} - \left( \frac{2(\hat{\rho}_1 - \hat{\rho}_0)}{\|\mu_0 - \mu_1\|_{\mathcal{H}}} \sqrt{\hat{\rho}_0} \right)^{1/2} \left( \frac{b_T}{T} \right)^{1/4}
$$

is usually better than that of the original estimator and the max-gap estimator. The drawbacks of bias correction are that it requires specific assumptions on $P_u$, and it is more computationally expensive than the max-gap estimator.

## C   Technical Proofs

**Notations.** For brevity of notation, we write $Y_{i,T}$ as $y_i$, $\| \cdot \|_{\mathcal{H}} = \| \cdot \|$, and $\langle \cdot, \cdot \rangle_{\mathcal{H}} = \langle \cdot, \cdot \rangle$. Denote

$$
\Delta(u, v) = \int_0^v \phi(Y_t(w)) dw - v/u \int_0^u \phi(Y_t(w)) dw,
$$

$$
\widehat{\Delta}(u, v) = 1/T \sum_{t=1}^{\lceil vT \rceil} \phi(Y_{t,T}) - v/(uT) \sum_{t=1}^{\lceil uT \rceil} \phi(Y_{t,T}).
$$

For a set of constants $a_T, b_T$, we write $a_T \lesssim b_T$ if there exist constants $C \in (0, +\infty), t_0 \in \mathbb{Z}_+$ such that $a_T \leq Cb_T$ for all $T \geq t_0$. Define

$$
V_1 = \int_0^{\rho^*} \mathbb{I}\left( L_0(u) > ub_T \right) du, \quad V_2 = \int_{\rho^*}^1 \mathbb{I}(T^{1/2} L_1(u) + T\mathcal{D}(u) \leq ub_T) du. \tag{23}
$$

## C.1 Technical Lemmas

First we will introduce some useful lemmas.

**Lemma 1.** *Suppose Assumption 1, 2, 3, 4 hold, if $b_T \to +\infty$, $b_T/T \to 0$, we have*

$$\mathbb{E}V_1 = \Theta\left(P(b_T)\exp\left\{-2b_T/\lambda_1\right\}\right),$$

*where $P(b_T)$ is a known function having polynomially bounded growth as $b_T \to \infty$.*

**Lemma 2.** *Under Assumptions 1, 2, 3, 4, if $b_T$ satisfies $b_T \to \infty$ and $b_T/T \to 0$ as $T \to +\infty$, when $T$ is sufficiently large, we have*

$$\left(\frac{\rho^* b_T}{mT}\right)^{1/\kappa} + o((\log T/T)^{1/\kappa}) \leq \mathbb{E}V_2$$

$$\leq \left(\frac{b_T}{mT}\left[\sqrt{\rho^* + \sigma_{\max}^2/\lambda_1} + \sqrt{\sigma_{\max}^2/\lambda_1}\right]^2\right)^{1/\kappa} + o((\log T/T)^{1/\kappa}),$$

*where $\sigma_{\max}^2 = \sup_{l,u} Var\left(\phi_l(Y_t(u))\right)$. Notice that we have $\sigma_{\max}^2 \geq \lambda_1$.*

## C.2 Derivation for Equation (10)

Denote

$$X_u(t) = \left(W(t) - t/uW(u)\right)/(\sqrt{u}/2).$$

with $W(\cdot)$ the standard Brownian motion.

Obviously $X_u$ is a centered non-stationary Gaussian process with almost surely continuous sample path. And the variance of $X_u$ attains its maximum over $[0, u]$ at the unique point $t = u/2 \in (0, u)$, and the maximum variance equals 1. Further, we have

$$\sigma(t) = \sqrt{\text{Var}(X_u(t))} = 1 - \frac{2}{u^2}\left|t - \frac{u}{2}\right|^2(1 + o(1)), \quad t \to \frac{u}{2},$$

$$r(s,t) = \text{Corr}\left(X_u(s), X_u(t)\right) = 1 - \frac{2}{u}|t - s|(1 + o(1)), \quad s, t \to \frac{u}{2},$$

$$\mathbb{E}\left[X_u(t) - X_u(s)\right]^2 \leq \frac{16}{u}|t - s|, \quad \forall s, t \in [0, u].$$

From Theorem 2.1 of Liu and Ji (2014), we have for $\lambda_1 \geq \lambda_2 \geq \cdots$ and $n$ mutually independent Brownian motion $W_l$'s:

$$\mathbb{P}\left(\max_{v \in (0,u)} \sum_{l=1}^{n} \lambda_l\left[\left(W_l(v) - \frac{v}{u}W_l(u)\right)^2\right] > x\right)$$

$$= \mathbb{P}\left(\max_{v \in (0,u)} \sum_{l=1}^{n} \lambda_l/\lambda_1\left[\left(W_l(v) - \frac{v}{u}W_l(u)\right)^2/(u/4)\right] > 4x/(u\lambda_1)\right)$$

$$= \mathbb{P}\left(\max_{v \in (0,u)} \sum_{l=1}^{n} \lambda_l/\lambda_1[X_u(v)]^2 > 4x/(u\lambda_1)\right)$$

$$= \frac{2^{1-q/2}}{\Gamma(q/2)}\left(\frac{4x}{u\lambda_1}\right)^{q/2-1/2}\exp\left\{-\frac{2x}{u\lambda_1}\right\}\mathcal{M}_{1,2,2/u,2/u^2}\prod_{l=1}^{n}(1 - \lambda_l/\lambda_1)^{-1/2}(1 + o(1)), \quad (24)$$

where $q$ is the multiplicity of $\lambda_1$ and $\mathcal{M}_{1,2,2/u,2/u^2} = 2\sqrt{2\pi}Pic_1$ with $Pic_1$ the Pickands constant defined as

$$Pic_1 = \lim_{s \to \infty}\frac{1}{S}\mathbb{E}\left(\exp\left\{\sup_{t \in [0,S]}\left(\sqrt{2}W(t) - t\right)\right\}\right).$$

It is known that $Pic_1 = 1$. Replacing $q$ and $\lambda_l$'s by their estimates, we obtain Equation (10).

## C.3  Proof of Theorem 4.4

*Proof.* **When** $u \in (0, \rho^*]$. From Theorem 16 of Tewes (2017), we have

$$\left( \frac{1}{\sqrt{T}} \sum_{i=1}^{\lceil Tu \rceil} (\phi(y_i) - \mathbb{E}\phi(Y)) \right)_{u \in [0, \rho^*]} \xrightarrow{w} (\boldsymbol{W}(u))_{u \in [0, \rho^*]},$$

where $\boldsymbol{W}(\cdot)$ is a Brownian motion in $\mathcal{H}$ and $\boldsymbol{W}(1)$ has the covariance operator $\Sigma : \mathcal{H} \to \mathcal{H}$, defined by

$$\langle \Sigma\phi(y), \phi(y') \rangle = \mathbb{E}_{y''} \left[ \langle \phi(y'') - \mathbb{E}\phi(y''), \phi(y) \rangle \langle \phi(y'') - \mathbb{E}\phi(y''), \phi(y') \rangle \right], \quad \forall y, y' \in \mathcal{H}.$$

From Equation (7) in the main paper, we have under the null,

$$\langle \Sigma\phi(y), \phi(y') \rangle = \mathbb{E}_{y''} \sum_{l,m} \phi_l(y)\phi_m(y') \left[ \phi_l(y'')\phi_m(y'') \right] = \sum_l \lambda_l \phi_l(y)\phi_l(y'),$$

as long as the last quantity is well-defined. Thus, we know $\boldsymbol{W}(\rho) = \left( \sqrt{\lambda_1} W_1(\rho), \sqrt{\lambda_2} W_2(\rho), \cdots \right)^\top$ where $W_l(\rho)$ and $W_m(\rho)$ are independent, standard Brownian motions if $l \neq m$. Thus, we have

$$T\widehat{\mathcal{D}}_T(u) = \left\| \frac{\lceil Tu \rceil - \lceil Tv \rceil}{\lceil Tu \rceil} \frac{1}{\sqrt{T}} \sum_{i=1}^{\lceil Tv \rceil} \phi(y_i) - \frac{\lceil Tv \rceil}{\lceil Tu \rceil} \frac{1}{\sqrt{T}} \sum_{i=\lceil Tv \rceil+1}^{\lceil Tu \rceil} \phi(y_i) \right\|^2$$

$$\xrightarrow{w} \left\| \boldsymbol{W}(v) - \frac{v}{u}\boldsymbol{W}(u) \right\|^2 = \sum_{l=1}^{\infty} \lambda_l \left( W_l(v) - \frac{v}{u}W_l(u) \right)^2,$$

and combined with the fact that $\mathcal{D}(u) = 0$ for anay $u \in [0, \rho^*]$, we get that Equation (16) holds.

**When** $u \in (\rho^*, 1]$. This part of proof follows from Proof of Theorem 2.1 in Dehling et al. (2017). First, we have the decomposition

$$T\widehat{\mathcal{D}}_T(u) = \left\| \frac{\lceil Tu \rceil - \lceil Tv \rceil}{\lceil Tu \rceil} \frac{1}{\sqrt{T}} \sum_{i=1}^{\lceil Tv \rceil} (\phi(y_i) - \mathbb{E}\phi(y_i)) - \frac{\lceil Tv \rceil}{\lceil Tu \rceil} \frac{1}{\sqrt{T}} \sum_{i=\lceil Tv \rceil+1}^{\lceil Tu \rceil} (\phi(y_i) - \mathbb{E}\phi(y_i)) \right.$$

$$\left. + \frac{\lceil Tu \rceil - \lceil Tv \rceil}{\lceil Tu \rceil} \frac{1}{\sqrt{T}} \sum_{i=1}^{\lceil Tv \rceil} \mathbb{E}\phi(y_i) - \frac{\lceil Tv \rceil}{\lceil Tu \rceil} \frac{1}{\sqrt{T}} \sum_{i=\lceil Tv \rceil+1}^{\lceil Tu \rceil} \mathbb{E}\phi(y_i) \right\|^2$$

$$= \left\| U_1 + \sqrt{T}U_2 \right\|^2 = \|U_1\|^2 + T\|U_2\|^2 + 2\sqrt{T}\langle U_1, U_2 \rangle, \tag{25}$$

where

$$U_1 = \frac{\lceil Tu \rceil - \lceil Tv \rceil}{\lceil Tu \rceil} \frac{1}{\sqrt{T}} \sum_{i=1}^{\lceil Tv \rceil} (\phi(y_i) - \mathbb{E}\phi(y_i)) - \frac{\lceil Tv \rceil}{\lceil Tu \rceil} \frac{1}{\sqrt{T}} \sum_{i=\lceil Tv \rceil+1}^{\lceil Tu \rceil} (\phi(y_i) - \mathbb{E}\phi(y_i)),$$

$$U_2 = \frac{\lceil Tu \rceil - \lceil Tv \rceil}{\lceil Tu \rceil} \frac{1}{T} \sum_{i=1}^{\lceil Tv \rceil} \mathbb{E}\phi(y_i) - \frac{\lceil Tv \rceil}{\lceil Tu \rceil} \frac{1}{T} \sum_{i=\lceil Tv \rceil+1}^{\lceil Tu \rceil} \mathbb{E}\phi(y_i).$$

Notice that the independence of $y_i$ and Assumption 1 implies Theorem 3 in Brown et al. (1971) holds, which says

$$\frac{1}{\sqrt{T}} \sum_{i=1}^{\lceil Tv \rceil} (\phi_l(y_i) - \mathbb{E}\phi_l(y_i)) \xrightarrow{w} G_l(v),$$

where $G_l$ is a Gaussian process. Moreover, for any $l \neq l'$,

$$\text{Cov}\left(G_l(v_1), G_{l'}(v_2)\right) = \lim_{T \to \infty} \mathbb{E}\left[\frac{1}{T} \sum_{i=1}^{\lceil Tv_1 \rceil} \left(\phi_l(y_i) - \mathbb{E}\phi_l(y_i)\right) \sum_{j=1}^{\lceil Tv_2 \rceil} \left(\phi_{l'}(y_j) - \mathbb{E}\phi_{l'}(y_j)\right)\right]$$

$$= \lim_{T \to \infty} \frac{1}{T} \sum_{i=1}^{\lceil Tv_1 \rceil} \mathbb{E}\left[\left(\phi_l(y_i) - \mathbb{E}\phi_l(y_i)\right)\left(\phi_{l'}(y_i) - \mathbb{E}\phi_{l'}(y_i)\right)\right]$$

$$+ \lim_{T \to \infty} \frac{1}{T} \sum_{i=1}^{\lceil Tv_1 \rceil} \sum_{j \neq i}^{\lceil Tv_2 \rceil} \mathbb{E}\left[\left(\phi_l(y_i) - \mathbb{E}\phi_l(y_i)\right)\left(\phi_{l'}(y_j) - \mathbb{E}\phi_{l'}(y_j)\right)\right] = 0$$

Thus, for different $l$, $G_l$'s are mutually independent.

So by definition of $U_1$, we have

$$U_{1,l} = \frac{\lceil Tu \rceil - \lceil Tv \rceil}{\lceil Tu \rceil} \frac{1}{\sqrt{T}} \sum_{i=1}^{\lceil Tv \rceil} \left(\phi_l(y_i) - \mathbb{E}\phi_l(y_i)\right) - \frac{\lceil Tv \rceil}{\lceil Tu \rceil} \frac{1}{\sqrt{T}} \sum_{i=\lceil Tv \rceil + 1}^{\lceil Tu \rceil} \left(\phi_l(y_i) - \mathbb{E}\phi_l(y_i)\right)$$

$$\xrightarrow{w} G_l(v) - v/u G_l(u). \tag{26}$$

Since

$$\|U_2\| \leq \|U_2 - \Delta(u, v)\| + \left\|\sqrt{T}\Delta(u, v)\right\|, \tag{27}$$

we have

$$\|U_2 - \Delta(u, v)\|$$

$$\leq \left(1 - \frac{v}{u}\right) \left\|\frac{1}{T} \sum_{i=1}^{\lceil Tv \rceil} \mathbb{E}\phi(y_i) - \int_0^v \mu(w)dw\right\| + \frac{v}{u} \left\|\frac{1}{T} \sum_{i=\lceil Tv \rceil + 1}^{\lceil Tu \rceil} \mathbb{E}\phi(y_i) - \int_v^u \mu(w)dw\right\|$$

$$\leq \left(1 - \frac{v}{u}\right) \left\|\frac{1}{T} \sum_{i=1}^{\lceil Tv \rceil} \mathbb{E}\phi(y_i) - \frac{1}{T} \sum_{i=1}^{\lceil Tv \rceil} \mu(i/T)\right\| + \left(1 - \frac{v}{u}\right) \left\|\frac{1}{T} \sum_{i=1}^{\lceil Tv \rceil} \mu(i/T) - \int_0^v \mu(w)dw\right\|$$

$$+ \frac{v}{u} \left\|\frac{1}{T} \sum_{i=\lceil Tv \rceil + 1}^{\lceil Tu \rceil} \mathbb{E}\phi(y_i) - \frac{1}{T} \sum_{i=1}^{\lceil Tv \rceil} \mu(i/T)\right\| + \frac{v}{u} \left\|\frac{1}{T} \sum_{i=\lceil Tv \rceil + 1}^{\lceil Tu \rceil} \mu(i/T) - \int_v^u \mu(w)dw\right\|. \tag{28}$$

Notice that from the Endpoint Approximation Theorem,

$$\left\|\frac{1}{T} \sum_{i=1}^{\lceil Tv \rceil} \mu(i/T) - \int_0^v \mu(w)dw\right\| = \left[\sum_{l=1}^{\infty} \left|\frac{1}{T} \sum_{i=1}^{\lceil Tv \rceil} \mu_l(i/T) - \int_0^v \mu_l(w)dw\right|^2\right]^{1/2}$$

$$\leq \sum_{l=1}^{\infty} \left|\frac{1}{T} \sum_{i=1}^{\lceil Tv \rceil} \mu_l(i/T) - \int_0^v \mu_l(w)dw\right| \leq \frac{v^2 \sum_{l=1}^{\infty} \max_{w \in (0,v)} \mu_l'(w)}{2T} \overset{(a)}{\leq} \frac{C'}{T}, \tag{29}$$

where (a) follows from Assumption 3 (iii). Similarly,

$$\left\|\frac{1}{T} \sum_{i=\lceil Tv \rceil + 1}^{\lceil Tu \rceil} \mu_l(i/T) - \int_v^u \mu(w)dw\right\| \leq \frac{(u-v)^2 \sum_{l=1}^{\infty} \max_{w \in (u,v)} \mu_l'(w)}{2T} \leq \frac{C''}{T}, \tag{30}$$

and

$$\left\| \frac{1}{T} \sum_{i=1}^{\lceil Tv \rceil} \mathbb{E}\phi(y_i) - \frac{1}{T} \sum_{i=1}^{\lceil Tv \rceil} \mu_l(i/T) \right\| = \left[ \sum_{l=1}^{\infty} \left| \mathbb{E}\frac{1}{T} \sum_{i=1}^{\lceil Tv \rceil} [\phi_l(y_i) - \phi_l(y_i(i/T))] \right|^2 \right]^{1/2}$$

$$\leq \left[ \sum_{l=1}^{\infty} \mathbb{E}\left[ \frac{1}{T} \sum_{i=1}^{\lceil Tv \rceil} [\phi_l(y_i) - \phi_l(y_i(i/T))] \right]^2 \right]^{1/2} = \left[ \mathbb{E}\left\| \frac{1}{T} \sum_{i=1}^{\lceil Tv \rceil} [\phi(y_i) - \phi(y_i(i/T))] \right\|^2 \right]^{1/2}$$

$$\overset{(a)}{\leq} \left[ \mathbb{E}\frac{v}{T} \sum_{i=1}^{\lceil Tv \rceil} \|\phi(y_i) - \phi(y_i(i/T))\|^2 \right]^{1/2} \overset{(b)}{\leq} \left[ \mathbb{E}\frac{v}{T} \sum_{i=1}^{\lceil Tv \rceil} C_1 \|y_i - y_i(i/T)\|^2 \right]^{1/2}$$

$$\overset{(c)}{\leq} \left[ v^2 C_1 \frac{1}{T^2} \mathbb{E}[U_{i,T}(i/T)]^2 \right]^{1/2} \overset{(d)}{\leq} \frac{v}{T}\sqrt{C_1 C}, \tag{31}$$

where $(a)$ uses Cauchy-Schwarz Inequality,$(b)$ follows from Assumption 3 (i), and $(c)(d)$ follows from the definition of locally stationary process. Similarly we have

$$\left\| \frac{1}{T} \sum_{i=\lceil Tv \rceil+1}^{\lceil Tu \rceil} \mathbb{E}\phi(y_i) - \frac{1}{T} \sum_{i=\lceil Tv \rceil+1}^{\lceil Tu \rceil} \mu_l(i/T) \right\| \leq \frac{u-v}{T}\sqrt{C_1 C}. \tag{32}$$

Combing Equation (28), (29), (30), (31), (32), we have

$$\|U_2 - \Delta(u,v)\| \leq \frac{C_3}{T}.$$

Thus,

$$U_2 \xrightarrow{w} \Delta(u,v).$$

Notice that

$$\|\Delta(u,v)\|^2 = \mathcal{K}(u,v),$$

thus we have

$$\sqrt{T}\left( \|U_2\|^2 - \mathcal{K}(u,v) \right) \leq \sqrt{T}\left[ \|U_2 - \Delta(u,v)\| + \|\Delta(u,v)\| \right]^2 - \sqrt{T}\mathcal{K}(u,v)$$

$$\leq \sqrt{T}\left[ \|U_2 - \Delta(u,v)\|^2 + \|\Delta(u,v)\|^2 \right.$$

$$\left. +2\|U_2 - \Delta(u,v)\| \|\Delta(u,v)\| \right] - \sqrt{T}\mathcal{K}(u,v)$$

$$\leq \sqrt{T}\left[ \frac{C_3^2}{T^2} + 2\frac{C_3}{T}\sqrt{\mathcal{K}(u,v)} \right], \tag{33}$$

and

$$\sqrt{T}\left( \|U_2\|^2 - \mathcal{K}(u,v) \right) \geq \sqrt{T}\left[ \|U_2 - \Delta(u,v)\| - \|\Delta(u,v)\| \right]^2 - \sqrt{T}\mathcal{K}(u,v)$$

$$\geq \sqrt{T}\left[ \|U_2 - \Delta(u,v)\|^2 + \|\Delta(u,v)\|^2 \right. \tag{34}$$

$$\left. -2\|U_2 - \Delta(u,v)\| \|\Delta(u,v)\| \right] - \sqrt{T}\mathcal{K}(u,v)$$

$$\geq -2\sqrt{T}\|U_2 - \Delta(u,v)\| \|\Delta(u,v)\|$$

$$\geq -2\sqrt{\mathcal{K}(u,v)}\frac{C_3}{T^{1/2}}. \tag{35}$$

Plugging Inequality (33), (35) into (25), we get

$$\sqrt{T}\left( \widehat{K}_T(u,v) - \mathcal{K}(u,v) \right) = T^{-1/2}\|U_1\|^2 + \sqrt{T}\left( \|U_2\|^2 - \mathcal{K}(u,v) \right) + 2\langle U_1, U_2 \rangle$$

$$\leq T^{-1/2}\|U_1\|^2 + \sqrt{T}\left[ \frac{C_3^2}{T^2} + 2\frac{C_3}{T}\sqrt{\mathcal{K}(u,v)} \right] + 2\langle U_1, U_2 \rangle,$$

$$\sqrt{T}\left( \widehat{K}_T(u,v) - \mathcal{K}(u,v) \right) = T^{-1/2}\|U_1\|^2 + \sqrt{T}\left( \|U_2\|^2 - \mathcal{K}(u,v) \right) + 2\langle U_1, U_2 \rangle$$

$$\geq T^{-1/2}\|U_1\|^2 - 2\sqrt{\mathcal{K}(u,v)}\frac{C_3}{T^{1/2}} + 2\langle U_1, U_2 \rangle.$$

Then in view of (26), we get

$$\sqrt{T}\left(\widehat{K}_T(u,v) - \mathcal{K}(u,v)\right) \xrightarrow{w} 2\sum_{l=1}^{\infty}\left(G_l(v) - v/uG_l(u)\right)\Delta_l(u,v) =: G(v,u), \qquad (36)$$

where the right hand side is well defined because for each $v, u$, it is a normal random variable with mean 0 and variance

$$\text{Var}(G(v,u)) = 4\sum_{l=1}^{\infty}\text{Var}\left(G_l(v) - v/uG_l(u)\right)\Delta_l^2(u,v)$$

$$\leq C_4\sum_{l=1}^{\infty}\Delta_l^2(u,v) \leq C_4\mathcal{K}(u,v) \leq C_4\mathcal{D}(u). \qquad (37)$$

$\square$

## C.4   Proof of Corollary 4.1

*Proof.* Notice that for any fixed constant $C > 0$,

$$\mathbb{P}\left(T\widehat{\mathcal{D}}_T(u) > C\right) = \mathbb{P}\left(T^{1/2}\widehat{\mathcal{D}}_T(u) > CT^{-1/2}\right).$$

From Theorem 4.4, we know that

$$T^{1/2}\widehat{\mathcal{D}}_T(1) = \Theta(T^{1/2}) + O_p(1).$$

Thus, the desired conclusion follows directly. $\square$

## C.5   Proof of Theorem 4.8

*Proof.* Notice

$$|\mathbb{E}V_1 - \mathbb{E}V_2| \leq l_1(\rho^{\infty}) = \mathbb{E}\,|V_1 - V_2| \leq \mathbb{E}V_1 + \mathbb{E}V_2.$$

Then Theorem 4.8 is a direct conclusion of Lemma 1 and 2, which show that the order of $\mathbb{E}V_1$ decreases when $b_T$ increases, while that of $\mathbb{E}V_2$ increases when $b_T$ increases. Thus, the optimal $b_T$ which minimizes $l_1^{\infty}$ should satisfy $\mathbb{E}V_1 = \Theta(\mathbb{E}V_2)$, which directly leads to Theorem 4.8. $\square$

## C.6   Proof of Lemma 1

*Proof.* On one hand, for any $u \in (0,1)$, we have from Lemma 1 of Inglot and Ledwina (2006) that

$$\mathbb{P}\left(\max_{v \in (0,u)}\sum_{l=1}^{\infty}\lambda_l\left[\left(W_l(v) - \frac{v}{u}W_l(u)\right)^2\right] > c_T(u)\right)$$

$$\geq \mathbb{P}\left(\sum_{l=1}^{q}\lambda_l\left[\left(W_l(\frac{u}{2}) - \frac{1}{2}W_l(u)\right)^2\right] > c_T(u)\right)$$

$$\geq \mathbb{P}\left(\chi_q^2 > \frac{4c_T(u)}{u\lambda_1}\right) \geq C_1\left[\frac{c_T(u)}{u}\right]^{q/2-1}\exp\left\{-\frac{2c_T(u)}{u\lambda_1}\right\}, \qquad (38)$$

where $q$ is the multiplicity of $\lambda_1$.

On the other hand, for any $u \in (0,1)$, from Chernoff bound, for any positive constant $C_2$, we have

$$\mathbb{P}\left(\max_{v \in (0,u)}\sum_{l=1}^{\infty}\lambda_l\left[\left(W_l(v) - \frac{v}{u}W_l(u)\right)^2\right] > c_T(u)\right)$$

$$\leq \mathbb{E}\exp\left\{-C_2c_T(u) + C_2\max_{v \in (0,u)}\sum_{l=1}^{\infty}\lambda_l\left[\left(W_l(v) - \frac{v}{u}W_l(u)\right)^2\right]\right\}$$

$$\overset{(a)}{=} e^{-C_2c_T(u)}\prod_{l=q+1}^{\infty}\mathbb{E}\exp\left\{C_2\lambda_l\max_{v \in (0,u)}\left(W_l(v) - \frac{v}{u}W_l(u)\right)^2\right\}. \qquad (39)$$

where $(a)$ follows from the fact that $W_l(\cdot)$'s are mutually independent processes. We know that for all $a > 0$,

$$\mathbb{P}\left(\max_{v\in(0,u)}\left(W_l(v) - \frac{v}{u}W_l(u)\right) \geq a\right) = \exp\left\{-\frac{2a^2}{u}\right\}.$$

We write $Z_l(u) = \exp\left\{C_2\lambda_l \max_{v\in(0,u)}\left(W_l(v) - \frac{v}{u}W_l(u)\right)^2\right\}$ and $\lambda_{q+1}/\lambda_1 = \xi \in (0,1)$.

We take $C_2 = \frac{2-\varepsilon_T}{u\lambda_1}$ with $\varepsilon_T = 1/b_T$. For all $l > q$, we have

$$\mathbb{E}Z_l(u) = 1 + \int_1^\infty \left[\mathbb{P}\left(Z_l(u) > a\right)\right] da = 1 + \int_1^\infty \left[\mathbb{P}\left(\max_{v\in(0,u)}\left(W_l(v) - \frac{v}{u}W_l(u)\right)^2 > \frac{\log a}{C_2\lambda_l}\right)\right] da \tag{40}$$

$$\leq 1 + 2\int_1^\infty \left[\mathbb{P}\left(\max_{v\in(0,u)}\left(W_l(v) - \frac{v}{u}W_l(u)\right) > \sqrt{\frac{\log a}{C_2\lambda_l}}\right)\right] da$$

$$\leq 1 + 2\int_1^\infty \exp\left\{-\frac{2\log a}{uC_2\lambda_l}\right\} da = 1 + 2\int_1^\infty a^{-2/(uC_2\lambda_l)} da = 1 + 2\frac{1}{2/(uC_2\lambda_l) - 1}$$

$$= 1 + \frac{2(2-\varepsilon_T)\lambda_l}{(2\lambda_1 - (2-\varepsilon_T)\lambda_l)} \leq 1 + \frac{\lambda_l}{\lambda_1}\frac{2(2-\varepsilon_T)}{(2-(2-\varepsilon_T)\xi)} < 1 + \frac{\lambda_l}{\lambda_1}\frac{2}{1-\xi}. \tag{41}$$

Notice that

$$\prod_{l=q+1}^\infty \mathbb{E}Z_l = \prod_{l=q+1}^\infty \left[1 + \frac{2\lambda_l}{\lambda_1(1-\xi)}\right] < +\infty, \tag{42}$$

is always satisfied since

$$\sum_{l=q+1}^\infty \log\left[1 + \frac{2\lambda_l}{\lambda_1(1-\xi)}\right] < \sum_{l=q+1}^\infty \frac{2\lambda_l}{\lambda_1(1-\xi)} < \frac{2\mathbb{E}k_0(y,y)}{\lambda_1(1-\xi)} < +\infty.$$

Similar to (41), we can prove that when $T$ is sufficiently large, for all $l \leq q$,

$$\mathbb{E}Z_l(u) < 1 + \frac{2(2-\varepsilon_T)}{\varepsilon_T} = 1 + 2b_T(2 - 1/b_T) < 5b_T. \tag{43}$$

Plugging (42) and (43) into Equation (39), we have

$$\mathbb{P}\left(\max_{v\in(0,u)}\sum_{l=1}^\infty \lambda_l\left[\left(W_l(v) - \frac{v}{u}W_l(u)\right)^2\right] > c_T(u)\right)$$

$$\lesssim (b_T)^q \exp\left\{-\frac{(2-1/b_T)c_T(u)}{u\lambda_1}\right\} \lesssim \left(\frac{c_T(u)}{u}\right)^q \exp\left\{-\frac{2c_T(u)}{u\lambda_1}\right\}. \tag{44}$$

Combining Inequality (38) and (44), for any fixed $u$, by definition of $V_1$, we have

$$\left[\frac{c_T(u)}{u}\right]^{q/2-1} \exp\left\{-\frac{2c_T(u)}{u\lambda_1}\right\} \lesssim \mathbb{E}V_1 \lesssim \left[\frac{c_T(u)}{u}\right]^q \exp\left\{-\frac{2c_T(u)}{u\lambda_1}\right\}.$$

$$\square$$

### C.7 Proof of Lemma 2

*Proof.* Notice that

$$\mathbb{E}V_2 = \int_{\rho^*}^1 \mathbb{P}\left(T^{1/2}L_1(u) + T\mathcal{D}(u) \leq ub_T\right) du.$$

**Upper bound for** $\mathbb{E}V_2$. Recall that Equation (37) says

$$\mathrm{Var}\left(G(v,u)\right) \le C_4 \mathcal{D}(u).$$

Then, $\forall v_0 \in \arg\max_{v\in[0,u]} \mathcal{K}(u,v)$, define

$$C_{3,l} := \mathrm{Var}\left(G_l(v_0) - \frac{v_0}{u}G_l(u)\right) \le \frac{\sigma_{\max}^2}{4} =: C_3, \quad \forall l \in \mathbb{Z}_+.$$

Then, from Section C.3, we know

$$\mathbb{P}\left(T^{1/2}L_1(u) + T\mathcal{D}(u) \le ub_T\right)$$

$$=\mathbb{P}\left(\sqrt{T}\max_{v\in(0,u)} 2\sum_{l=1}^{\infty}\left(G_l(v) - \frac{v}{u}G_l(u)\right)\Delta_l(u,v) \le c_T(u) - T\mathcal{D}(u)\right)$$

$$\le\mathbb{P}\left(2\sum_l \Delta_l(v_0,u)\left(G_l(v_0) - \frac{v_0}{u}G_l(u)\right) \le \frac{c_T(u)}{\sqrt{T}} - \sqrt{T}\mathcal{D}(u)\right)$$

$$\le\mathbb{P}\left(N\left(0, 4\sum_l \Delta_l^2(v_0,u)\mathrm{Var}\left(G_l(v_0) - \frac{v_0}{u}G_l(u)\right)\right) \le \frac{c_T(u)}{\sqrt{T}} - \sqrt{T}\mathcal{D}(u)\right)$$

$$\le\mathbb{P}\left(N\left(0, 4\sum_l \Delta_l^2(v_0,u)C_{3,l}\right) \le \frac{c_T(u)}{\sqrt{T}} - \sqrt{T}\mathcal{D}(u)\right)$$

$$=\mathbb{P}\left(N(0,1) \le \frac{c_T(u)/\sqrt{T} - \sqrt{T}\mathcal{D}(u)}{\sqrt{4\sum_l C_{3,l}\Delta_l^2(v_0,u)}}\right).$$

For any $\varepsilon > 0$, when $(u - \rho_0)^{\kappa} \le \frac{c_T(u)}{(1-2\varepsilon)mT}$,

$$\mathbb{P}\left(N(0,1) \le \frac{c_T(u)/\sqrt{T} - \sqrt{T}\mathcal{D}(u)}{\sqrt{4\sum_l C_{3,l}\Delta_l^2(v_0,u)}}\right) \le 1.$$

When $(u - \rho_0)^{\kappa} > \frac{ub_T}{(1-2\varepsilon)mT}$: from Assumption 4 and $b_T/T \to 0$, we know that for any positive constant $\delta > 0$, when $T$ is sufficiently large,

$$\frac{c_T(u)}{\sqrt{T}} - \sqrt{T}\mathcal{D}(u) < 0, \quad \mathcal{D}(u) \ge (1 - \delta\varepsilon)m(u - \rho_0)^{\kappa} \ge \frac{1 - \delta\varepsilon}{1 - 2\varepsilon}\frac{c_T(u)}{T},$$

and thus,

$$\mathbb{P}\left(N(0,1) \le \frac{c_T(u)/\sqrt{T} - \sqrt{T}\mathcal{D}(u)}{\sqrt{4\sum_l C_{3,l}\Delta_l^2(v_0,u)}}\right)$$

$$\le\mathbb{P}\left(N(0,1) \le \frac{c_T(u)/\sqrt{T} - \sqrt{T}\mathcal{D}(u)}{\sqrt{4C_3\mathcal{D}(u)}}\right)$$

$$\le\mathbb{P}\left(N(0,1) \le -(2-\delta)\varepsilon\sqrt{\frac{c_T(u)}{4(1-\delta\varepsilon)(1-2\varepsilon)C_3}}\right)$$

$$\le\frac{C_5}{\sqrt{b_T}}\exp\left\{-\frac{(2-\delta)^2\varepsilon^2 ub_T}{8(1-\delta\varepsilon)(1-2\varepsilon)C_3}\right\}. \tag{45}$$

Denote

$$a = \exp\left\{-\frac{(2-\delta)^2\varepsilon^2 b_T}{8(1-\delta\varepsilon)(1-2\varepsilon)C_3}\right\}.$$

Combing the cases where $(u - \rho_0)^\kappa \leq \frac{c_T(u)}{(1-2\varepsilon)mT}$ and $(u - \rho_0)^\kappa > \frac{c_T(u)}{(1-2\varepsilon)mT}$, and denote $\rho_0 = \rho^*$, we have when $T$ is sufficiently large,

$$
\begin{aligned}
\mathbb{E}V_2 &\leq \int_{\rho_0}^1 \frac{C_5}{\sqrt{b_T}} \exp\left\{-\frac{(2-\delta)^2\varepsilon^2 u b_T}{8(1-\delta\varepsilon)(1-2\varepsilon)C_3}\right\} du + (1+\varepsilon)\left(\frac{\rho_0 b_T}{(1-2\varepsilon)mT}\right)^{1/\kappa} \\
&= \frac{C_5}{\sqrt{b_T}} \frac{a^{\rho_0}(1-a^{1-\rho_0})}{\log[1/a]} + (1+\varepsilon)\left(\frac{\rho_0 b_T}{(1-2\varepsilon)mT}\right)^{1/\kappa} \\
&\leq C_6 T^{-(2-\delta)^2\varepsilon^2\lambda_1\rho_0/[16\kappa(1-\delta\varepsilon)(1-2\varepsilon)C_3]} + (1+\varepsilon)\left(\frac{\rho_0 b_T}{(1-2\varepsilon)mT}\right)^{1/\kappa}. 
\end{aligned} \tag{46}
$$

We choose $\varepsilon$ as the solution to
$$
\frac{(2-\delta)^2\varepsilon^2\lambda_1\rho_0}{16(1-\delta\varepsilon)(1-2\varepsilon)C_3} = 1.
$$

Thus we get

$$
\varepsilon = \varepsilon_0 = \frac{-4C_3 + \sqrt{16C_3^2 + 4\lambda_1\rho_0 C_3}}{\lambda_1\rho_0} = \frac{C_3}{C_3 + \sqrt{C_3^2 + \lambda_1\rho_0 C_3/4}} \in (0, 0.5).
$$

Plugging into (46) and let $\varepsilon \to 0$, we have

$$
\mathbb{E}V_2 \leq o\left(\left(\frac{b_T}{T}\right)^{1/\kappa}\right) + \left(\frac{b_T}{mT}\left[\sqrt{\frac{\sigma_{\max}^2}{\lambda_1}} + \sqrt{\frac{\sigma_{\max}^2}{\lambda_1} + \rho_0}\right]^2\right)^{1/\kappa}. \tag{47}
$$

**Lower bound for $\mathbb{E}V_2$.** For any positive constant $\varepsilon$, when $(u - \rho_0)^\kappa (1 + 2\varepsilon) \geq \frac{c_T(u)}{Tm}$, we have
$$
\mathbb{P}\left(T^{1/2}L_1(u) + T\mathcal{D}(u) \leq c_T(u)\right) \geq 0.
$$

When $(u - \rho_0)^\kappa (1 + 2\varepsilon) < \frac{c_T(u)}{Tm}$, write $C_9 = \rho_0 + \left(\frac{\rho_0 b_T}{(1+2\varepsilon)Tm}\right)^{1/\kappa}$, when $u > \rho_0$ is sufficiently close to $\rho_0$, we have
$$
\frac{\mathcal{D}(u)}{(u - \rho_0)^\kappa} \leq (1 + \varepsilon)m.
$$

Thus, when $T$ is sufficiently large,

$$
\begin{aligned}
&\mathbb{P}\left(T^{1/2}L_1(u) + T\mathcal{D}(u) \leq c_T(u)\right) \\
&= \mathbb{P}\left(\sqrt{T} \max_{v\in(0,u)} 2\sum_{l=1}^\infty \left(G_l(v) - \frac{v}{u}G_l(u)\right)\Delta_l(u, v) \leq c_T(u) - T\mathcal{D}(u)\right) \\
&= \mathbb{P}\left(\sqrt{T} \max_{v\in(0,u)} G(v, u) \leq c_T(u) - T\mathcal{D}(u)\right) \\
&= \mathbb{P}\left(\frac{\max_{v\in(0,u)} G(v, u)}{\sqrt{\mathcal{D}(u)}} \leq \frac{c_T(u) - T\mathcal{D}(u)}{\sqrt{T\mathcal{D}(u)}}\right) \\
&\geq \mathbb{P}\left(\frac{\max_{v\in(0,u)} G(v, u)}{\sqrt{\mathcal{D}(u)}} \leq \frac{c_T(u) - (1+\varepsilon)Tm(u-\rho_0)^\kappa}{\sqrt{(1+\varepsilon)Tm(u-\rho_0)^\kappa}}\right) \\
&\geq \mathbb{P}\left(\frac{\max_{v\in(0,u)} G(v, u)}{\sqrt{\mathcal{D}(u)}} \leq \varepsilon\sqrt{\frac{c_T(u)}{(1+\varepsilon)(1+2\varepsilon)}}\right) \\
&\geq \inf_{u\in(\rho_0, C_9]} \mathbb{P}\left(\frac{\max_{v\in(0,u)} G(v, u)}{\sqrt{\mathcal{D}(u)}} \leq \varepsilon\sqrt{\frac{c_T(u)}{(1+\varepsilon)(1+2\varepsilon)}}\right) \\
&\overset{(a)}{=} 1 - o(1) \geq 1 - \varepsilon,
\end{aligned}
$$

where $G(v, u)$ is defined in (36) and (a) follows from the fact that $b_T \to \infty$ and Equation (37) which says that

$$\frac{\max_{v \in [0,u]} G(v, u)}{\sqrt{\mathcal{D}(u)}} = O_p(1), \quad \Rightarrow \quad \mathbb{P}\left(\frac{\max_{v \in [0,u]} G(v, u)}{\sqrt{\mathcal{D}(u)}} > \varepsilon \sqrt{\frac{c_T(u)}{(1+\varepsilon)(1+2\varepsilon)}}\right) = o(1).$$

Thus, for any $\varepsilon \in (0, 1)$, we have

$$\mathbb{E}V_2 = \int_{\rho_0}^1 \mathbb{P}\left(T^{1/2}L_1(u) + T\mathcal{D}(u) \le c_T(u)\right) du \ge (1-\varepsilon)(C_9 - \rho_0) = (1-\varepsilon)\left(\frac{\rho_0 b_T}{(1+2\varepsilon)Tm}\right)^{1/\kappa}. \tag{48}$$

Combining (47) and (48) and let $\varepsilon \to 0$ in (48), we obtain the desired conclusion. $\qquad\square$

## C.8 Proof of Theorem 4.10

*Proof.* Let $f(\cdot) = \left|\int_{\rho^*}^1 \mathbb{I}(\cdot \le c_T(u)) \, du + \int_0^{\rho^*} [\mathbb{I}(\cdot \le c_T(u)) - 1] \, du\right|$. We know that $f$ is a bounded, continuous function when $c_T(\cdot)$ is continuous. Since $l_1(\cdot) = \mathbb{E}f(\cdot)$, from Portmanteau Theorem, we have

$$l_1(\rho^\infty) - l_1(\hat{\rho}) \to 0, \quad \text{as } T \to \infty.$$

Similarly,

$$l_1(\rho_\infty) - l_1(\check{\rho}) \to 0, \quad \text{as } T \to \infty.$$

Then, Theorem 4.10 is a direct consequence of Theorem B.1. $\qquad\square$

## C.9 Proof of Theorem B.1

*Proof.* Notice that

$$\mathbb{E}[\rho^\infty - \rho^*]_+ \le \mathbb{E}V_2, \quad \mathbb{E}[\rho^* - \rho^\infty]_+ \le \mathbb{E}V_1,$$

where $V_1, V_2$ are defined in Equation (23). Thus, the bound on $\mathbb{E}[\rho^\infty - \rho^*]_+, \mathbb{E}[\rho^* - \rho^\infty]_+$ is a direct consequence of Lemma 1 and Lemma 2.

Thus, we only need to prove the bounds for $[\rho_\infty - \rho^*]_+$ and $[\rho^* - \rho_\infty]_+$. The key is that the basic inequality always holds:

$$(\rho_\infty - \rho^*) b_T \ge T\mathcal{D}(\rho_\infty) + w(\rho_\infty)L(\rho_\infty) - T\mathcal{D}(\rho^*) - w(\rho^*)L(\rho^*).$$

**Overestimation.** For any $\varepsilon \in (0, 1/2)$ and $x > \left(\frac{2(\rho^*+x)b_T}{(1-\varepsilon)mT}\right)^{1/\kappa}$, we have

$$\mathbb{P}(\rho_\infty > \rho^* + x)$$
$$= \mathbb{P}\left((\rho_\infty - \rho^*) b_T \ge T\mathcal{D}(\rho_\infty) + T^{1/2}L_1(\rho_\infty) - L_0(\rho^*), \rho_\infty > \rho^* + x\right)$$
$$= \mathbb{P}(\rho_\infty > \rho^* + x) - \mathbb{P}\left((\rho_\infty - \rho^*) b_T < T\mathcal{D}(\rho_\infty) + T^{1/2}L_1(\rho_\infty) - L_0(\rho^*), \rho_\infty > \rho^* + x\right)$$
$$\le \mathbb{P}(\rho_\infty > \rho^* + x) - \mathbb{P}\left(\left\{\rho_\infty b_T < T\mathcal{D}(\rho_\infty) + T^{1/2}L_1(\rho_\infty), \rho^* b_T > L_0(\rho^*)\right\} \cap \{\rho_\infty > \rho^* + x\}\right)$$
$$\le \mathbb{P}\left(\left\{\rho_\infty b_T \ge T\mathcal{D}(\rho_\infty) + T^{1/2}L_1(\rho_\infty) \quad \text{or} \quad \rho^* b_T \le L_0(\rho^*)\right\} \cap \{\rho_\infty > \rho^* + x\}\right)$$
$$\le \mathbb{P}\left(\rho_\infty b_T \ge T\mathcal{D}(\rho_\infty) + T^{1/2}L_1(\rho_\infty), \rho_\infty > \rho^* + x\right) + \mathbb{P}(\rho^* b_T \le L_0(\rho^*))$$
$$\overset{(a)}{\le} \mathbb{P}\left(\rho_\infty b_T \ge T\mathcal{D}(\rho_\infty) + T^{1/2}L_1(\rho_\infty), \rho_\infty > \rho^* + x\right) + Cb_T^{q/2-1} \exp\left\{-\frac{2b_T}{\lambda_1}\right\}$$
$$\overset{(b)}{\le} \frac{C_5}{\sqrt{b_T}} \exp\left\{-\frac{\varepsilon^2 \rho^* b_T}{8(1-\varepsilon)(1-2\varepsilon)C_3}\right\} + Cb_T^{q/2-1} \exp\left\{-\frac{2b_T}{\lambda_1}\right\}.$$

where $(a)$ follows from Equation (38), and $(b)$ follows from taking $\delta = 1$ in Equation (45) of Section C.7.

Let $x = \left(\frac{4b_T}{mT}\right)^{1/\kappa}$. Obviously for $T$ sufficiently large, there exists some $\varepsilon_0 \in (0, 1/2)$ such that $x > \left(\frac{2(\rho^* + x)b_T}{(1-\varepsilon)mT}\right)^{1/\kappa}$ is satisfied. Then we have

$$\mathbb{P}\left(\rho_\infty > \rho^* + \left(\frac{4b_T}{mT}\right)^{1/\kappa}\right) \to 0.$$

Then, we have for any $x > 0$,

$$\mathbb{P}\left(\rho_\infty > \rho^* + x\right)$$
$$= \mathbb{P}\left((\rho_\infty - \rho^*)\,b_T \geq T\mathcal{D}(\rho_\infty) + T^{1/2}L_1(\rho_\infty) - L_0(\rho^*), \rho_\infty > \rho^* + x\right)$$
$$= \mathbb{P}\left(L_1(\rho_\infty) - T^{-1/2}L_0(\rho^*) \leq T^{-1/2}(\rho_\infty - \rho^*)b_T - T^{1/2}\mathcal{D}(\rho_\infty), \rho_\infty > \rho^* + x,\right.$$
$$\left.\rho_\infty < \rho^* + (4b_T/(mT))^{1/\kappa}\right) + \mathbb{P}\left(\rho_\infty \geq \rho^* + (4b_T/(mT))^{1/\kappa}\right)$$
$$\leq \mathbb{P}\left(L_1(\rho_\infty) - T^{-1/2}L_0(\rho^*) \leq (4b_T/(mT))^{1/\kappa}\,T^{-1/2}b_T - T^{1/2}\mathcal{D}(\rho_\infty), \rho_\infty > \rho^* + x\right) + o(1)$$
$$= \mathbb{P}\left(\frac{L_1(\rho_\infty) - T^{-1/2}L_0(\rho^*)}{\sqrt{\mathcal{D}(\rho_\infty)}} \leq T^{1/2}\left[\frac{C}{\sqrt{\mathcal{D}(\rho_\infty)}}\left(\frac{b_T}{T}\right)^{1/\kappa+1} - \sqrt{\mathcal{D}(\rho_\infty)}\right], \rho_\infty > \rho^* + x\right) + o(1).$$

Let

$$x = \left(\frac{b_T}{mT\log b_T}\right)^{1/\kappa}.$$

From Assumption 4, we have for all $\rho_\infty > \rho^* + x$ and $T$ sufficiently large,

$$\mathcal{D}(\rho_\infty) \geq \mathcal{D}(\rho^* + x) \geq \frac{b_T}{2T\log b_T}.$$

Thus, as $T \to \infty$,

$$T^{1/2}\left[\frac{C}{\sqrt{\mathcal{D}(\rho_\infty)}}\left(\frac{b_T}{T}\right)^{1/\kappa+1} - \sqrt{\mathcal{D}(\rho_\infty)}\right] \to -\infty.$$

Since

$$\frac{L_1(\rho_\infty) - T^{-1/2}L_0(\rho^*)}{\sqrt{\mathcal{D}(\rho_\infty)}} = O_p(1),$$

we have

$$\mathbb{P}\left(\frac{L_1(\rho_\infty) - T^{-1/2}L_0(\rho^*)}{\sqrt{\mathcal{D}(\rho_\infty)}} \leq T^{1/2}\left[\frac{C}{\sqrt{\mathcal{D}(\rho_\infty)}}\left(\frac{b_T}{T}\right)^{1/\kappa+1} - \sqrt{\mathcal{D}(\rho_\infty)}\right], \rho_\infty > \rho^* + x\right) \to 0,$$

and thus,

$$[\rho_\infty - \rho^*]_+ = O_p\left(\left(\frac{b_T}{T\log b_T}\right)^{1/\kappa}\right) = o_p\left(\left(\frac{b_T}{T}\right)^{1/\kappa}\right).$$

**Underestimation.** On the other hand, from Markov Inequality, notice $L_0 > 0$, we have

$$\mathbb{P}\left(\rho^* - \rho_\infty > x\right) \leq \mathbb{P}\left(xb_T < (\rho^* - \rho_\infty)\,b_T \leq -L_0(\rho_\infty) + L_0(\rho^*)\right)$$
$$\leq \mathbb{P}\left(xb_T < L_0(\rho_\infty) + L_0(\rho^*)\right)$$
$$\leq \frac{\mathbb{E}\left[L_0(\rho_\infty) + L_0(\rho^*)\right]}{b_Tx} = \frac{(\rho^* + \rho_\infty)C}{b_Tx}.$$

Thus,

$$[\rho^* - \rho_\infty]_+ = O_p\left(1/b_T\right).$$

$\square$

## C.10  Proof of Theorem B.3

**Upper Bound for Delay.** Recall that

$$\mathbb{E}\left[\rho^\infty - \rho_0\right]_+ = \mathbb{E}\left[V_2 - V_1\right]_+ \le \mathbb{E}V_2,$$

From Lemma 2, we know that when $T$ is sufficiently large,

$$\mathbb{E}\left[\rho^\infty - \rho_0\right]_+ \le o\left(\left(\frac{b_T}{T}\right)^{1/\kappa}\right) + \left(\frac{b_T}{mT}\left[\sqrt{\frac{\sigma_{\max}^2}{\lambda_1}} + \sqrt{\frac{\sigma_{\max}^2}{\lambda_1} + \rho_0}\right]^2\right)^{1/\kappa}.$$

where $\kappa$ and $m$ depends on the actual change.

**Lower Bound for Delay.** Using Jensen's Inequality, we have

$$\mathbb{E}\left[\rho^\infty - \rho_0\right]_+ = \mathbb{E}\left[V_2 - V_1\right]_+ \ge \left[\mathbb{E}V_2 - \mathbb{E}V_1\right]_+ .$$

When $b_T > \frac{\lambda_1 \log T}{2\kappa}$, Lemma 1 implies $\mathbb{E}V_1 = o\left(\left(\frac{b_T}{T}\right)^{1/\kappa}\right)$. Combined with Lemma 2, we have

$$\mathbb{E}\left[\rho^\infty - \rho_0\right]_+ \ge \left[\mathbb{E}V_2 - \mathbb{E}V_1\right]_+ \ge \left(\frac{\rho_0 b_T}{Tm}\right)^{1/\kappa} + o\left(\left(\frac{b_T}{T}\right)^{1/\kappa}\right).$$

## C.11  Proof of Corollary B.1

*Proof.* In mixture setting where Equation (21) holds, we have

$$\kappa = 4, \quad m = \frac{\|\mu_0 - \mu_1\|^2}{4(\rho_1 - \rho_0)^2}.$$

In the abrupt case, we have

$$\kappa = 2, \quad m = \frac{\|\mu_0 - \mu_1\|^2}{2}.$$

Thus, we can directly obtain Corollary B.1 by plugging the value of $\kappa$ and $m$ into Theorem B.3.  □

## D  Experimental Details and Additional Results

**Baseline setup.** For $Z_w$, we construct the binary graph using the 5-MST as suggested by Chu et al. (2019) and after construction, it is implemented by the R package gSeg (Chen et al., 2014). For $\hat{\rho}^{\mathrm{mix}}$, we use the Cramer-von Mises measure $\Phi^{\mathrm{CvM}}$ defined in Section 5.1 of Quessy (2019) because empirical studies in Quessy (2019) shows that $\Phi^{\mathrm{CvM}}$ slightly outperforms the other two measures it proposed.

For $Z_w$ and $Q$, we need to choose a distance measure $d$ (which generalizes the Euclidean distance used in Matteson and James (2014)). For KCpA and the proposed estimator $\hat{\rho}, \check{\rho}$, we need to choose a kernel $k$. For Vogt and Dette (2015), we need to choose a function class $\mathcal{F}$. The choice of $d$, $k$, $\mathcal{F}$ have critical influence on the performance of estimators. For fairness, we set them to be as similar to each other as possible. In location model, we set $\mathcal{F} = \{f : x \mapsto x_i, \forall i = 1, \cdots, d\}$, and $d(y_i, y_j) = \|y_i - y_j\|^2$ with $\|\cdot\|$ being Euclidean distance, and $k(y_i, y_j) = y_i^\top y_j$. In volatility model, we set $\mathcal{F} = \{f : x \mapsto x^2\}$, and $d(y_i, y_j) = (y_i^2 - y_j^2)^2$, and $k(y_i, y_j) = y_i^2 y_j^2$. In network model, we set $\mathcal{F} = \{f : x \mapsto x_{ij}, \forall i, j = 1, \cdots, 10\}$, $d(y_i, y_j) = \|y_i - y_j\|_F^2$ with $\|\cdot\|_F$ being Frobenius norm, and $k(y_i, y_j) = \mathrm{vec}(y_i)^\top \mathrm{vec}(y_j)$ with $\mathrm{vec}(x)$ being a vectorized version of $x$.

**Power comparison.** Notice that $\hat{\rho}^{\mathrm{poly}}, \hat{\rho}^{\mathrm{one\text{-}sided}}, \hat{\rho}^{\mathrm{gen}}$ do not consider detection, and $\hat{\rho}^{\mathrm{mix}}$ is computationally expensive with 500 permutations, and are thus excluded from power comparisons.

With the data generating process described in the main paper, most methods have an empirical power equal to 1, which makes the comparison of power not so informative. Thus, we focus on a more challenging setting with a modified data generating scheme. For location model (2), we include univariate cases with $\varepsilon_t \sim N(0, 1)$ and $\mu_1, \mu_2, \mu_3, \mu_4$ defined in the main paper replaced by $0.3\mu_1$,

Table 2: Comparison of empirical power averaged over 20 simulations. Numbers after $\pm$ is the standard deviation of the empirical power $\hat{p}$ calculated via $\hat{p}(1-\hat{p})/\sqrt{20}$.

| MODEL | LOCATION | | | | | | | VOLATILITY | | NETWORK |
|---|---|---|---|---|---|---|---|---|---|---|
| DIM CHANGE | 1 LINEAR | 1 QUADRATIC | 1 ONE-SIDED | 1 COMPLEX | 10 LINEAR | 20 LINEAR | 50 LINEAR | 1 LINEAR | 1 COMPLEX | $10^2$ LINEAR |
| $\hat{\rho}, \check{\rho}$ | **0.85±0.03** | **0.90±0.02** | **0.60±0.05** | **0.70±0.05** | **0.65±0.05** | **0.20±0.04** | **0.85±0.03** | **0.60±0.05** | **0.70±0.05** | **0.65±0.05** |
| $Q$ | 0.75±0.04 | **0.90±0.02** | 0.55±0.06 | 0.65±0.05 | 0.45±0.06 | 0.10±0.02 | 0.55±0.06 | 0.45±0.06 | **0.70±0.05** | 0.60±0.05 |
| KCpA | 0.75±0.04 | **0.90±0.02** | 0.55±0.06 | 0.65±0.05 | 0.45±0.06 | **0.20±0.04** | 0.45±0.06 | 0.45±0.06 | **0.70±0.05** | 0.35±0.05 |
| $Z_w$ | 0.15±0.03 | 0.10±0.02 | 0.20±0.04 | 0.05±0.01 | 0.15±0.03 | 0.10±0.02 | 0.00±0.00 | 0.00±0.00 | 0.00±0.00 | 0.05±0.01 |

Table 3: Type I error of proposed method estimated using Equation (10) averaged over 100 simulations, and under significance level $\alpha = 0.05$.

| DIMENSION | 1 | 5 | 10 | 20 | 50 |
|---|---|---|---|---|---|
| TYPE I ERROR | 0.06 | 0.06 | 0.09 | 0.34 | 0.91 |

$0.3\mu_2$, $0.2\mu_3$, and $0.1\mu_4$. For multivariate $Y_{t,T} \in \mathbb{R}^d$ where $\varepsilon_t \sim N_d(0, I_d)$: we replace $\mu_1$ by $0.1\mu_1$ for $d = 10$ and $0.07\mu_1$ for $d = 20, 50$. For volatility model (3), we replace $\sigma_1 = \mu_1 + 1, \sigma_4 = \mu_4 + 1$ by $\sigma_1 = 0.17\mu_1 + 1$ and $\sigma_4 = 0.08\mu_4 + 1$. For network model, we change the original $p(u)$ into $p(u) = 0.05\mathbb{I}(1/3 \le u \le 2/3)(3u - 1) + 0.05\mathbb{I}(u \ge 2/3) + 0.1$ with the other settings unchanged.

Results on power comparison are summarized in Table 2. We observe that power of the proposed method is slightly higher than that of $Q$ and KCpA. All three of them (the proposed method, $Q$ and KCpA) perform significantly better than $Z_w$, which is as expected because $Z_w$ uses the binary similarity graph to contruct statistics and is less informative than the other three.

**Type I error (calibration).** We evaluate the calibration of p-values using Equation (10). We generate multivariate normal random vectors from $N(0, I_p)$ where $p$ is the dimension. And we choose $d$ as squared Euclidean distance. Results are summarized in Table 3. We observe that for low dimensions $p \le 10$, the estimated type I error is close to significance level $\alpha = 0.05$; as dimension $p$ grows, however, p-value yielded from (10) severely under-estimates the true type I error. The root reason is that in our setting, as $p$ grows, the multiplicity $q$ of leading eigenvalue grows, and the error in the estimated leading eigenvalue also increases dramatically. Thus, under cases where $q$ might be large, we suggest using permutation tests instead to estimate p-value.

**String data.** One advantage of the proposed kernel-based method is its ability to handle structured data. Thus, here we look into its performance on string data. We generate data in this way: each $Y_{t,T}$ is a length-3 string with each character in string randomly sampled from some set with replacement. For $\rho^* \le 1/3$, this set is $\{$ "$a$", "$b$", "$c$" $\}$, and we denote the distribution of strings generated in this way by $F_0$; for $\rho^* \ge 2/3$, this set is $\{$ "$A$", "$B$", "$C$" $\}$, and similarly we define $F_1$; for $\rho \in (1/3, 2/3)$, each string is generated from a mixture of $F_0$ and $F_1$: $(2 - 3\rho)F_0 + (3\rho - 1)F_1$. Here we compare against KCpA and we set kernel to be $k(y, y') = \exp\{-\|y - y'\|^2/h\}$ where $\|\cdot\|$ denotes the edit (Levenshtein) distance (Levenshtein et al., 1966) between two strings. Again the bandwidth $h$ is tuned among $\{0.01, 0.05, 0.1, 1, 5, 10, 20, 50, 100, 500\}$. The results are shown in the last column of Table 4. We observe that again $\hat{\rho}, \check{\rho}$ are performing better than KCpA.

Table 4: Comparison of localization error averaged over 20 testing sets. Numbers after $\pm$ is the standard deviation of localization error. The choice of bandwidth $h$ for both methods are tuned on 20 independently generated tuning sets.

| METHOD | $l_1$ ERROR |
|---|---|
| $\hat{\rho}$ | **0.08±0.01** |
| $\check{\rho}$ | 0.09±0.01 |
| KCpA | 0.16±0.01 |

# E   More Related Work

**Trend filtering.**   Trend filtering was recently proposed by Kim et al. (2009) for nonparametric regression, and has been developed by Tibshirani (2014); Wang et al. (2015); Ramdas and Tibshirani (2016); Sadhanala and Tibshirani (2017); Moghtaderi et al. (2013). The goal of trend filtering is to estimate the trend of a time series while penalizing for changes in the absolute $k$-th order discrete derivatives over the input points. We note that it is a related, but different problem from what we consider in this work. One difference is that, in our setting, we deal with data with no trend, while trend filtering aims to estimate the trend as well as the changes in the trend. Another difference is that trend filtering was originally designed for scalar type of observations, and extension for other type of observations has to be developed case-by-case. In contrast, our framework naturally allows general types of observations.