# OpenReview forum: "A nonparametric method for gradual change problems with statistical guarantees"
_NeurIPS.cc/2021/Conference — NeurIPS 2021 Poster_

### Official Review · Reviewer_W1jH · 2021-07-11

**Rating:** 8
**Confidence:** 5

**Summary:**

This paper analyzes the problem of characterizing a gradual change in the probability measure of an independent-across-time time series $Y_t,$ $t \in \{0, 1, \ldots, T\}$ over an observed interval $[0, T]$. The authors are interested in:  1) detecting whether or not a change has occurred; 2) Localizing the change within $[0,T]$. The authors seek only some time $\rho^* \in [0,T]$ where the change begins to occur. They propose a procedure for detecting the change and localizing it, in addition to providing theoretical results that establish statistical guarantees, and experimental results that show the benefits of their approach.



**Ethical Concerns:**

No concerns.

**Limitations And Societal Impact:**

There are no limitations or negative societal impact implications.

**Main Review:**

Novelty: The paper formulates a novel gradual anomaly detection and localization problem that goes beyond the historied quickest change detection problem in which one seeks to detect and localize an abrupt change with probability one, e.g., Lorden 1971, as the number of observations tends to infinity. They characterize the solution to their gradual change problem under the assumptions that the gradual change must be continuous with respect to the standard topology in $[0,T]$ and the weak topology on the space of measures $\{P_t \}$. That is, as $t \rightarrow x$ in $[0,T]$, $P_t$ must approach $P_x$ weakly.  For localization, there is less novelty. The authors take the change-point estimate of (Vogt 2015) and modify it slightly with hyperparameter $c_T$. They then minimize the expected absolute difference of this change-point estimate minus the true change point with respect to the asymptotic ($T \rightarrow \infty$) distribution over possible values of $c_T$.

Significance:  The authors present a new change point detection statistic that is highly versatile and requires few assumptions on the distribution of the time-series in which the change point may occur. Moreover, the proposed statistic can detect a wide variety of changes. This contrasts with existing approaches that seek a specific change (requiring domain knowledge). The authors also prove asymptotic distributions of their statistic, find an optimal localization estimator of the CPD with respect to a heuristic established in (Vogt 2015), and evaluate p-values using the derived asymptotic distribution.

Scholarship: The authors do an excellent job at placing their contribution in the context of previous work. Existing approaches for this problem (Vogt 2015) partition the interval of observation into $[0, v] \cup (v, T]  = [0, T]$ and compare, as a function of $v$, the difference between $E[f(Y_t)]$ for $t \in [0, v]$ vs $t \in (v, T]$. This comparison is made for a set of functions $f \in \mathcal{F}$ that is pre-specified. The authors rightfully criticize this approach because the set of functions $\mathcal{F}$ is typically finite, which excludes some classes of changes in distribution. Instead of the statistic of (Vogt 2015), which returns the max-max over partitions $v$ and functions $f$ that yield the largest difference in $E[f(Y_t)]$,  the authors' define a new statistic,
equation (6), that measures the similarity in measure of the $Y$'s across the partitions, comparing it to the similarity within  the partition. Intuitively, this statistic is 0 if there is no change in [0, T] because the ``within'' and ``across'' estimates are close. This is all clearly exposited in the paper.

Completeness of theory:  The authors show that the distribution of their statistic under the no-change hypothesis is asymptotically Chi-squared, allowing them to calculate $p-$values required to defined the stopping time of their procedure. They then argue that their approach encapsulates infinite-cardinality function sets $\mathcal{F}$ of the (Vogt 2015) approach with the appropriate choice of kernel function.

Experiments: The empirical results section is very complete. There are several comparisons to existing methods for both detection and localization of change-points. The proposed algorithm is compared to previous approaches on simulated data involving various types of gradual changes for which the previous approaches were designed and  should achieve a high level of performance in localization. The proposed algorithm is also compared to 3 abrupt change-point detection methods, showing that the proposed algorithm achieves exceptional performance on detection and good performance on localization. Although, as mentioned earlier, it is being compared to localization estimators that are designed to detect a specific type of change.

Quality of writing: The quality of writing is excellent overall. The only suggestion is that the authors should be more specific about which limit is in question when using the word "asymptotic." This was not entirely clear in the body of the paper, and required me to look to the Theory section and appendix.

Additional comments:

Although not stated in the document, the hypothesis testing problem can be stated as a test of hypotheses of the distribution change function $f$: H0: $f$ is constant on $[0, T]$ vs.
H1: $f$ is continuous with respect to the weak topology, but not constant on $[0, T]$. This would better contrast the proposed problem against Lorden's approach in which $f$ is a Heaviside step function on $[0, \infty)$ under H1 and constant under H0.  Stating the problem in this way clarify the difference between the standard formulation of the change detection problem and the authors' proposed formulation.



**Time Spent Reviewing:**

3 hours

---

> ### Author Response · Authors · 2021-08-10
> **Author Response to Reviewer W1jH**
>
> We thank the reviewer for the very detailed review, the encouraging comments and constructive feedback!
>
> * About the ambiguous use of the term ‘asymptotic’: Thank you for pointing this out! We were aiming at basic intuitions and motivations in Section 3, so we were not very strict in some terms. We appreciate the careful reading of the Theory and Appendix. We will make the statements clearer in Section 3 in the revised manuscript.
>
> * About the alternative formulation of the hypothesis testing problem: we consider the suggestion from the reviewer to be an excellent idea, and it helps contrast with the abrupt setting. We will adopt it in the revised manuscript, plus an additional statement to characterize the point $\rho^*$ where $f$ starts to change.

---

> ### Comment · Reviewer_W1jH · 2021-08-26
> **Response to authors**
>
> I have read the author response and my score stands unchanged.

---

### Official Review · Reviewer_hFCk · 2021-07-12

**Rating:** 5
**Confidence:** 4

**Summary:**

The manuscript proposed a kernel-based nonparametric method to detect and localize gradual change points in time-ordered observations. Theoretical properties are studied and the proposed method is assessed by several simulation examples and two real data applications.

**Limitations And Societal Impact:**

Besides the major comments listed in the Main Review section, I list some minor comments as follows:

(1) Several citation format errors. For example "Vogt et al. (2012)" should be "Vogt  (2012)", and "Vogt et al. (2015)" should be "Vogt and Dette (2015)".

(2) The definition of the alternative hypothesis on page 2 (between lines 48 and 49) is a bit confusing. As the authors allow u to approach \rho^* from above and \epsilon to approach 0 from above.  I wonder if it is feasible to detect the alternative in an extreme case when the change is allowed to happen at any slow rate.

(3) In line 96 of page 4, u can reach the left boundary 0 where equation (6) is not well defined.

(4) In line 87 of page 3, what does the term "non-Euclidean data" mean here?

**Main Review:**

Originality and Significance: The manuscript follows the gradual changes detection framework introduced in Vogt and Dette (2015). The authors proposed to change the sets of moment functions used in Vogt and Dette (2015) to a kernel function. In my regard, this manuscript provides marginal contributions to existing work.

Quality and Clarity: The presentation of the paper needs to be improved. (1) The claim that the proposed new statistic is applicable to any data type and any generating process may be exaggerated. The authors assume the observed sequence is independent, local stationery, and of gradual changes. These assumptions certainly rule out some data-generating processes from the discussion. (2) The authors claimed they do not put any assumptions on the data type of distribution of Y_{t,T}. However, it is not clear to me if the proposed method is appropriate if Y_{t,T} is categorical as the distances among the levels may not be well defined.  (3) The max-gap estimator proposed in the paper is also a bit unclear to me. If there are multiple change points, will the monotonic pattern shown in Figure 3 always hold? (4) The presentation of Table 1 needs to be improved. The simulations are summarized over only 20 replications. (5) Both real datasets studied has clear serial dependence which may not be the best showcase examples as the theory is developed under independence.



**Time Spent Reviewing:**

48

---

> ### Author Response · Authors · 2021-08-10
> **Author Response to Reviewer hFCk**
>
> We thank the reviewer for the detailed comments! We hope that our responses clarify the issues the reviewer raised, issues that affected the rating received by the paper.
>
> * About improvement over existing work [1]:
>
>     We politely disagree that this manuscript is only a marginal improvement over existing work. The framework in [1] can be summarized as: (i) finding a test statistic $D$ that satisfies certain conditions, (ii) using the estimator $\hat\rho^{\text{gen}}$ (between line 135 and 136 in this manuscript) for localization.  We agree that we are following this framework, with improvements in:
> (i) proposing new test statistic $D$;
> (ii) proposing a modified max-gap estimator $\check\rho$ (Equation (14)) ;
> (iii) adding a detection step.
>
>     The biggest advancement lies in (i). Also, improvements in (ii)-(iii) are evident but less important. Let us discuss (i) in greater detail.
>
>     1. Difference between the proposed test statistics and the existing one: The proposed statistics are Equation (5)(6), and the existing one is in Equation (4). The former is based on kernels, while the existing one is based on a finite set of pre-specified functions. Both their formats and building elements are quite different.
> The decomposition (Equation (8)) is perhaps more similar to existing statistics (Equation (4)). However, we note that Equation (8) is not our final statistics. We derive it purely for the purpose of understanding the property of proposed statistics and comparing against existing ones. With that in mind, we try to maximize its similarities with Equation (4) by expressing it using elements that appear in Equation (4). We do not judge the similarity of our statistics against existing ones based on that.
>
>     2. Similarity between the proposed procedure and the existing one:
> We agree that the proposed statistics (Equation (5)(6)) and the existing one (Equation (4)) are similar in that they both take the form $max_{v\in[0,u]}$. And the proposed localization estimator (12) is similar to the existing one (between line 135 and 136) in that they both take the form of averages of indicator functions.  However, as far as we know, replacing the key statistics in an existing framework by a new one is an acknowledged contribution in change point literature. For example, in the problem of abrupt change point detection, many existing methods use a test statistic of the form $max_{u\in[0,1]}D(u)$, and the change point is estimated as $\hat\rho=\arg\max_{u\in[0,1]}D(u)$. Their only difference lies in a new statistic $D(\cdot)$ (e.g., [2][3][4][5], just to name a few). We refer the reviewer to a review of change point literature [6], which clearly groups most existing change point methods into common frameworks, with variation only in the specific statistic.
>
> * About the claim that new statistic is applicable to any data type and any generating process:
>
>     1. On the generating process: We are grateful for pointing this out! We apologize for any ambiguities in that statement. We were trying to emphasize the relaxed assumptions on the marginal distribution of each $Y_{t,T}$, instead of the whole time series. We will make this statement more strict and less-ambiguous in the revised manuscript.
>
>     2. On data types:  Theoretically, the proposed statistic can be applied to any data type that has a valid kernel. This holds even for the categorical variables mentioned by the reviewer. For example, one can use various methods to learn an embedding for each categorical variable. Then, the kernel can be defined based on the learned embedding vector. This manuscript decouples the question of defining a good kernel (for special data types) from the proposed procedure to give it more generality. We believe that kernel choice for special data type is an interesting and important direction for future work.
>
> * About the behavior of max-gap estimator in multiple change points:
> The short answer is “yes”, it is still monotonic in the ‘local’ area around the first change point. This can be roughly seen from the population version of $D(u)$ (defined in Equation (15)), which, according to Assumption 4, is 0 on $[0,\rho^*]$, but is strictly positive and monotonic on $(\rho^*,\rho^*+\varepsilon]$ for some $\varepsilon>0$. Whether there are multiple change points, or how the distribution changes does not affect this conclusion. But more strictly speaking, the definition of ‘multiple change point’ itself is ambiguous in our setting. Currently, we don’t put any assumptions on the distribution of $Y_{t,T}$ after the change point. It means that we allow the distribution to be **always** changing after the change point $\rho^*$. In such cases, it is hard to define multiple change points.
>
> * About the presentation of Table 1:
> We have also reported the standard deviation of each result in Table 1. As we can see, the standard deviation is small relative to the difference among estimators. So we only report the result based on 20 experiments in the current manuscript. If the reviewer requires it, we can increase the number if experiments in the revised manuscript.
>
> * About serial dependence in real dataset:
> Thank you for pointing out this issue! It is a common practice in statistical literature to make simplifying assumptions (e.g., independence, abrupt changes rather than gradual) for sake of theoretical derivations, while applying methods on real datasets that possibly do not satisfy the constraint (e.g., [2-5]). We follow this practice here due to two considerations: (a) it is difficult to verify whether a certain dataset satisfies all required conditions; (b) it is important to show that the proposed method can work well even if some assumptions are not satisfied strictly.
>
> * On minor comments:
>
>     1. about citation format errors: thanks for pointing that out! We will modify them in the revised manuscript.
>
>     2. About the question ‘whether it is feasible to detect the alternative in an extreme case when the change is allowed to happen at any slow rate’: We are not sure what you refer to by ‘change is allowed to happen at any slow rate’. In this manuscript, we require $\varepsilon$ to be fixed. The limit case you mentioned can be incorporated into Corollary 4.1, where we allow the magnitude of change point measured in $\mathcal{D}(1)$ to approach 0 at certain rates. When $\varepsilon$ approaches 0 from above (and assume that $\varepsilon$ cannot be increased), it means the only segment that contains the change $(\rho^*,\rho^*+\varepsilon]$ is increasingly shorter and shorter. In this case, there is a certain threshold such that if $\varepsilon$ is smaller than that threshold, we cannot detect the change point. From Corollary 4.1, that order for the boundary threshold should be $T^{-1/4}$.
>
>     3. About un-well-defined Equation (6) when u approach the left boundary 0: Thank you for pointing it out! We will modify it to $u\in(0,1]$ in the revised manuscript.
>
>     4. About the term non-Euclidean: We are sorry for the ambiguity. For example, contrary to scalars and multivariate vectors which are Euclidean, networks, pictures, texts are considered non-Euclidean. We will make it more unambiguous in the revised manuscript.
>
> We hope that these comments clarify some of the issues the reviewer raised, issues that affected the rating received by the paper.
>
> References:
>
> [1] Vogt, Michael, and Holger Dette. "Detecting gradual changes in locally stationary processes." The Annals of Statistics 43.2 (2015): 713-740.
>
> [2] Chen, Hao, and Nancy Zhang. "Graph-based change-point detection." The Annals of Statistics 43.1 (2015): 139-176.
>
> [3] Chu, Lynna, and Hao Chen. "Asymptotic distribution-free change-point detection for multivariate and non-euclidean data." The Annals of Statistics 47.1 (2019): 382-414.
>
> [4] Matteson, David S., and Nicholas A. James. "A nonparametric approach for multiple change point analysis of multivariate data." Journal of the American Statistical Association 109.505 (2014): 334-345.
>
> [5] Dubey, Paromita, and Hans-Georg Müller. "Fréchet change-point detection." The Annals of Statistics 48.6 (2020): 3312-3335.
>
> [6] Truong, Charles, Laurent Oudre, and Nicolas Vayatis. "Selective review of offline change point detection methods." Signal Processing 167 (2020): 107299.

---

> ### Comment · Reviewer_hFCk · 2021-08-26
> **Response to authors**
>
> Thank you for your detailed response letter. It addresses some of my concerns and I now lean to accept this paper.

---

### Official Review · Reviewer_oZvV · 2021-07-12

**Rating:** 7
**Confidence:** 2

**Summary:**

This paper proposes a nonparametric, kernel-based approach for detecting and localizing gradual changes in time-series data, provides theoretical guarantees on its performance, and validates its efficacy via numerical experiments with both synthetic and real data.

**Main Review:**

I think this work makes solid contributions to the problem of detecting and localizing gradual changes.
* The proposed nonparametric algorithms are flexible and have few tuning parameters, requiring minimal assumption/knowledge on data distributions; the intuitions behind algorithm design are well explained.
* The presentation of theoretical results seem rigorous to me, with a complete set of technical assumptions and concise statements (as well as intuitive explanations) of the main results (though I did not carefully check the correctness). On the downside, the presented results are mostly asymptotic in nature.
* The numerical experiments are convincing; in particular, the authors are honest about when the proposed methods work better or worse than other existing algorithms.
* The comparison with related work is rather exhaustive.  In terms of writing, this paper is well written and easy to follow.

My only suggestion is to add a quick but formal explanation to Lines 93-94, i.e. why the defined "within-group similarity" should be large compared with "between-group similarity", where "similarity" is defined by kernel. This point might not be immediately clear to readers who are not familiar with change detection or RKHS theory. Such an explanation will also help to explain Remark 3.1, e.g. why dot-product kernel is suggested for location model, but might fail for other model of changes.

--------
After rebuttal:
Thanks for the authors' response. I decide to keep my score.

**Time Spent Reviewing:**

2

---

> ### Author Response · Authors · 2021-08-10
> **Author Response to Reviewer oZvV**
>
> We thank the reviewer for your comments and constructive suggestions!
>
> * About the asymptotic nature of presented results:
>
>     Thank you for raising this suggestion! In the current manuscript, there are two main reasons we did not pursue more accurate finite-sample theory:
>
>     1. Currently, all theory in this manuscript serves the purpose of defining the detection and localization steps (e.g., determining the critical value in detection step, determining the threshold $c_T(\cdot)$ in localization step). From this starting point, we find asymptotic results suffice for our need.
>
>     2. Establishing finite-sample bounds might require more strict assumptions on data distributions. As we try to minimize assumptions on data distributions, we did not make further attempts in this direction.
>
>     We are grateful for this comment;  it could be an interesting direction to utilize higher order distributional assumptions and derive corresponding finite-sample bounds in future work.
>
> * About  lines 93-94: Thank you for the constructive suggestion! We will add an explanation to lines 93-94, and link it to the choice of kernels in Remark 3.1.

---

### Official Review · Reviewer_ZPrG · 2021-07-16

**Rating:** 7
**Confidence:** 4

**Summary:**

The paper proposes a method for the detection and localization of a gradual change in the distribution of sequence data. In contrast to existing work (which mostly focuses on abrupt changes in the distribution), this paper considers a smooth, gradual change. The approach is non-parametric and does not make major assumptions on the data/change model. The paper introduces a kernel-based statistic to detect the change that captures similarities between the observations before and after the change has occurred.


**Main Review:**

+ This is a solid paper and is well written. It provides a good balance between theory and experimentation.
+ The problem statement dispenses with model assumptions made in prior work such as particular parametric models, changes in pre-specified moments of a distribution, monotonicity of the change, and mixture models.
+ The max-gap estimator of the change point is promising and has good empirical performance with finite samples.
+ The paper gives theoretical asymptotic guarantees.
- If I understand correctly, the approach is a special case of [Vogt et al., 2015]. In particular, by setting the function f to be the feature map associated with the kernel of a Reproducing Kernel Hilbert Space, we get the proposed statistic by invoking the kernel trick. So, the novelty beyond [Vogt et al. 2015] is somewhat limited.
- The paper assumes independent observations, which may be unrealistic for practical applications with a gradual change. The paper claims that their method can be adapted to the setting with correlated observations, but obviously this is not considered in this work.
- The assumptions in the theoretical analysis of Section 4 are a little obscure. The paper should do a better job motivating these assumptions.




**Time Spent Reviewing:**

4

---

> ### Author Response · Authors · 2021-08-10
> **Author Response to Reviewer ZPrG**
>
> We thank the reviewer for your comments and constructive feedback!
>
> * Concerns on the novelty beyond Vogt (2015) [1]:
>
>     Thank you for raising this concern! We do not believe that the proposed statistic is a special case of [1]. Actually, [1] is a special case of our proposed statistic. One big difference is that the statistic in [1] (although conceptually feasible) does not allow an infinite cardinality F because it cannot be computed due to the infiniteness. The deployment of kernels solves this issue. To notice that [1] is a special case of the proposed statistic, we can simply set the kernel $k(y,y’) = \sum_{f \in F}(f(y)-f(y’))^2$.
> Another difference is that our statistic is decomposed as max over sum of squares (Equation(8)), while that in [1] is decomposed as max over max (Equation (4)). This difference is important as it brings us a much simpler and tractable asymptotic distribution compared to that in [1].
>
>     We want to further clarify that our final statistic is defined in Equation (5)(6), which is sufficiently different from [1] (Equation (4)). Equation (8) is derived purely for the purpose of understanding the properties and comparing against [1].
>
> * About the independence assumptions:
>
>     Thank you for raising this issue! We agree that independence might be restrictive for some real data, yet it is a commonly used assumption in the change point literature when introducing new statistics (e.g. [2][3]), especially for nonparametric change point methods where correlations are more difficult to handle. We do not consider dependence in this manuscript due to the same considerations.
> The correlation will change the independence between the Wiener processes in Equation (16), but other forms of limiting distributions will not change. Consequently, the critical values for detection and threshold for localization will also change, and the determination of them will become more complicated. As the current manuscript aims at introducing new statistics, we did not further pursue a more complicated correlation structure. We will leave it as an important direction for future work. We are grateful for this suggestion.
>
> * About the assumptions in Section 4:
>
>     Thank you for the suggestion! We agree that clarifications will improve the manuscript, and we will add more intuitive examples to motivate the assumptions in the revised manuscript.
>
> References:
>
> [1] Vogt, M., & Dette, H. (2015). Detecting gradual changes in locally stationary processes. Annals of Statistics, 43(2), 713–740.
>
> [2] Chen, Hao, and Nancy Zhang. "Graph-based change-point detection." The Annals of Statistics 43.1 (2015): 139-176.
>
> [3] Chu, Lynna, and Hao Chen. "Asymptotic distribution-free change-point detection for multivariate and non-euclidean data." The Annals of Statistics 47.1 (2019): 382-414.

---

> > ### Comment · Reviewer_ZPrG · 2021-08-26
> > **Re: authors response**
> >
> > I have read the reviewers' comments and the authors' responses. I continue to believe this is a good and solid paper, and that the strengths outweigh the weaknesses. I recommend "accept" so I will keep my score unchanged.

---

### Official Review · Reviewer_5aeA · 2021-07-21

**Rating:** 7
**Confidence:** 3

**Summary:**

This paper considers the problem of detecting gradual change-points. This setting is more general than the abrupt change-point detection setting, and is often encountered in applied situations. The authors propose a kernel-based statistics to estimate a single change, and prove the asymptotic consistency of their estimator. The numerical experiment are convincing.

**Limitations And Societal Impact:**

There are no negative societal impact in my opinion.

**Main Review:**

This work builds on the setting and results of [1] and propose a (conceptually) similar estimator. However, the authors make an original contribution by resorting to a kernel to improve the estimator of [1]. Thanks to this, the change-point detection procedure can now be used without knowing a priori the class of transformations that best measure the change. In addition, the estimator is shown to be asymptotically consistent, making the complete procedure theoretically sound.


Overall, the paper is clear and well-written. Two minor improvements could be made in my opinion:
- Summarize somewhere (in a pseudo-code if space allows it, or in a few sentences) all the steps needed to estimate a change and compute the associated p-value. All parameters and equations are spread out within the paper, making it difficult for a motivated reader to implement the method. Also, the computational complexity of the procedure is an interesting piece of information to provide for readers.
- There lacks references to the Maximum Mean Discrepancy (MMD) which is a kernel-based statistical test and is also used to do change-point detection. At first sight, the estimator $\hat{\Kappa}_T$ (Equation 6) seems very close to the MMD between the group before $[Tv]$ and the group after $[Tv]$.

[1] Vogt, M., & Dette, H. (2015). Detecting gradual changes in locally stationary processes. Annals of Statistics, 43(2), 713–740.


**Time Spent Reviewing:**

4

---

> ### Author Response · Authors · 2021-08-10
> **Author Response to Reviewer 5aeA**
>
> We thank the reviewer for the useful comments and constructive suggestions!
>
> * About summary of all steps: we are grateful for the constructive suggestion and we agree that it will improve the understanding of the methods and its implementation! We will add an algorithm box to summarize all steps in the revised manuscript.
>
> * About computational complexity: thank you for pointing this out! The computational complexity of the procedure is $O(T^2)$ using an additional storage of $O(T^2)$ to store the cumulative kernel matrix $A = [a_{ij}]$ where $a_{ij}=\sum_{l=1}^i\sum_{m=1}^j k(Y_{l,T}, Y_{m, T})$. We will discuss this issue in a revised manuscript.
>
> * About references to MMD: Thank you for pointing it out! We will add a reference and detailed discussion to it in the revised manuscript. We agree that the estimator $\hat{\Kappa}_T$ (Equation (6)) is similar to, but different from, MMD. The main difference lies in the additional scaling factor $v^2(u-v)^2/u^2$, which is crucial for the success of the proposed procedure in gradual CPD. Without the scaling factor, $\hat {\mathcal{D}}_T(u)$ (Equation (5)) will not have a well-defined limiting distribution (due to the behavior of $\hat\Kappa_T(u,v)$ around $v=0$ and $v=u$). And consequently, the detection and localization step which are based on this limiting distribution becomes infeasible. For the original MMD which was initially proposed in [1] in two sample testing problems, this issue does not exist because the two samples to be tested are fixed (so we do not need to take the maximum over $v\in[0,u]$).
>
> References:
>
> [1] Gretton, Arthur, et al. "A kernel two-sample test." The Journal of Machine Learning Research 13.1 (2012): 723-773.

---

### Decision · Program_Chairs · 2021-09-28

**Decision:**

Accept (Poster)

**Comment:**

The authors are proposing a general method for detecting and localizing gradual changes without a specific data model, type or prior knowledge on the features. Despite these relaxations, they provide a new method with guarantees, which they empirically demonstrate. Overall, this is great scholarly work, from its writing to empirical demonstrations and the work should be accepted.

**Consistency Experiment:**

NeurIPS has a long history of experimentation. In 2014, NeurIPS ran an experiment in which 10% of submissions were reviewed by two independent committees to quantify the randomness in the review process. This year, we repeated a variant of this experiment to see how the quality of the review process has changed over time.  This paper was part of the experiment and was therefore assigned to two committees (consisting of reviewers, an Area Chair, and a Senior Area Chair) that reached independent decisions.  If both committees made the same recommendation, this recommendation was followed. If a single committee recommended acceptance, the paper was accepted (with the exception of a few cases in which the other committee identified what we considered a fatal flaw, e.g., an error in a key result).

This copy’s committee reached the following decision: **Accept (Spotlight)**

The other committee assigned to the paper recommended **Reject**.  You can find the other set of reviews, along with any follow up discussion with the authors here:
https://openreview.net/forum?id=zwkj1_pxFM